# Descriptive comparison of admission characteristics between pandemic waves and multivariable analysis of the association of the Alpha variant (B.1.1.7 lineage) of SARS-CoV-2 with disease severity in inner London

Luke B Snell [ID],[1,2] Wenjuan Wang [ID],[3] Adela Alcolea-Medina,[1,4]
Themoula Charalampous [ID],[1] Rahul Batra,[1] Leonardo de Jongh,[5] Finola Higgins,[5]
Gaia Nebbia [ID],[1,2] COG-UK HOCI Investigators, Yanzhong Wang [ID],[3]
Jonathan Edgeworth,[1,2] Vasa Curcin[3]

LBS and WW contributed equally.

LBS and WW are joint first authors.
JE and VC are joint senior authors.

For numbered affiliations see end of article.

**Correspondence to**
Dr Luke B Snell;
luke.snell@nhs.net

## ABSTRACT

**Background** The Alpha variant (B.1.1.7 lineage) of SARS-CoV-2 emerged and became the dominant circulating variant in the UK in late 2020. Current literature is unclear on whether the Alpha variant is associated with increased severity. We linked clinical data with viral genome sequence data to compare admitted cases between SARS-CoV-2 waves in London and to investigate the association between the Alpha variant and the severity of disease.

**Methods** Clinical, demographic, laboratory and viral sequence data from electronic health record systems were collected for all cases with a positive SARS-CoV-2 RNA test between 13 March 2020 and 17 February 2021 in a multisite London healthcare institution. Multivariate analysis using logistic regression assessed risk factors for severity as defined by hypoxia at admission.

**Results** There were 5810 SARS-CoV-2 RNA-positive cases of which 2341 were admitted (838 in wave 1 and 1503 in wave 2). Both waves had a temporally aligned rise in nosocomial cases (96 in wave 1 and 137 in wave 2). The Alpha variant was first identified on 15 November 2020 and increased rapidly to comprise 400/472 (85%) of sequenced isolates from admitted cases in wave 2. A multivariate analysis identified risk factors for severity on admission, such as age (OR 1.02, 95% CI 1.01 to 1.03, for every year older; p<0.001), obesity (OR 1.70, 95% CI 1.28 to 2.26; p<0.001) and infection with the Alpha variant (OR 1.68, 95% CI 1.26 to 2.24; p<0.001).

**Conclusions** Our analysis is the first in hospitalised cohorts to show increased severity of disease associated with the Alpha variant. The number of nosocomial cases was similar in both waves despite the introduction of many infection control interventions before wave 2.

## Strengths and limitations of this study

► Published evidence on whether the Alpha variant of SARS-CoV-2 causes more severe disease (COVID-19) is mixed.

► Our study benefits from a long study window, including patients since the beginning of the SARS-CoV-2 pandemic.

► Our outcome measure for severity, hypoxia on admission, reflects the natural history of disease prior to medical intervention and hospital treatment.

► Our analysis adjusts for comorbidities, a feature missing from many of the population-level studies currently published.

## BACKGROUND

SARS-CoV-2 infection has led to the death of over 4 million individuals worldwide since its emergence in China during December 2019, with over 120 000 deaths reported in the UK as of July 2021. In London, the estimated incidence of new cases in the first wave peaked around 23 March 2020 at 2.2%[1] and then rapidly declined following non-pharmacological interventions. Hospital admissions peaked about 1 week later,[2] reflecting the median period of symptoms before hospital presentation. A 'second wave' of infections started in London around the beginning of October 2020.[3]

Genome sequencing identified the Alpha variant (the B.1.1.7 lineage) around the South East England, which spread rapidly as part of the emerging second wave.[4] This occurred prior to widespread vaccination, with only 25% of the adult population receiving the first dose by mid-February 2021.[5] The Alpha variant has been associated with increased transmissibility in community studies,[6 7] and community studies associate the variant with

increased mortality.[8 9] However, published studies in hospitalised patients suggested no increase in need for ventilation or mortality.[10]

Changes in transmissibility and severity have the potential to affect the burden on healthcare systems, and modify the characteristics of cases presenting to hospitals, including the demographics, comorbidities and severity of disease associated with SARS-CoV-2 infection.

## Objectives

We linked clinical datasets with local SARS-CoV-2 variant analysis to compare admission characteristics of hospitalised cases during the two waves of infection and to look at the association of the Alpha variant with severity of disease at presentation to the hospital.

## METHODS

### Setting

Guy's and St Thomas' NHS Foundation Trust (GSTT) is a multisite healthcare institution providing general and emergency services predominantly to the South London boroughs of Lambeth and Southwark. An acute-admitting site (St Thomas' Hospital) has an adult emergency department, with a large critical care service including one of the UK's eight nationally commissioned extracorporeal membrane oxygenation (ECMO) centres for severe respiratory failure. A second site (Guy's Hospital) provides more inpatient services such as elective surgery, cancer care and other specialist services. A paediatric hospital (Evelina London) acts as a general and specialised referral centre. Several satellite sites for specialist services like dialysis, rehabilitation and long-term care are also part of the institution. GSTT receives patients from regional hospitals predominantly critical care through 'mutual aid' schemes.

### SARS-CoV-2 laboratory testing

Our laboratory began testing on 13 March 2020 with initial capacity for around 150 PCR tests per day, before increasing to around 500 tests per day in late April during wave 1 and up to 1000 tests per day during wave 2 (online supplemental figure 1).

Testing commenced during the first wave on 13 March 2020 was limited to cases requiring admission or inpatients who had symptoms of fever or cough, as per national recommendation; guidance suggested cases which did not require admission should not be tested. For wave 2, all cases admitted to the hospital were screened and underwent universal interval screening at varying time points. Staff testing for symptomatic healthcare workers (HCWs) was also introduced towards the end of wave 1. Comparative analysis was therefore restricted to SARS-CoV-2 RNA-positive cases requiring admission. Cases without laboratory confirmation of SARS-CoV-2 infection were not included.

Assays used for the detection of SARS-CoV-2 RNA include PCR testing using Aus Diagnostics or by the Hologic Aptima SARS-CoV-2 Assay. Nucleic acid was first extracted using the QIAGEN QIAsymphony SP system and a QIAsymphony DSP Virus/Pathogen Mini Kit (catalogue number 937036) with the off-board lysis protocol.

### Definitions and participants

Cases were identified by the first positive SARS-CoV-2 RNA test. Cases were placed in mutually exclusive categories with the following definitions: (1) outpatients; (2) testing through occupational health; (3) emergency department (ED) attenders not subsequently admitted within 14 days; (4) patients admitted within 14 days of a positive test; (5) nosocomial cases, defined based on European Centre for Disease Prevention and Control (ECDC) definitions, as those having a first positive test on day 8 or later after admission to the hospital where COVID-19 was not suspected on admission;[11] and (6) interhospital transfers.

For the purpose of comparison, only the inpatient group admitted within 14 days following a positive test was taken forward for onward comparison. This methodology of only including admissions was adopted to prevent increased testing during the pandemic affecting case ascertainment and biasing severity of cases. This is evidenced in online supplemental figure 1, with tests increasing steadily from 100 per day to more than 1000 per day. Additionally, in wave 2, more interhospital transfers of severe cases requiring ECMO were received, mostly several days after admission. This category of patients was therefore excluded from analysis to prevent biasing towards severe disease.

A composite data point for 'hypoxia' was created, equivalent to WHO ordinal scale of $\geq 4$,[12] with cases taken to be hypoxic if on admission they had oxygen saturations of <94%, if they were recorded as requiring supplemental oxygen or if the fraction of inspired oxygen was recorded as being greater than 0.21.

### Determination of SARS-CoV-2 lineage

Whole-genome sequencing of residual samples from SARS-CoV-2 cases was performed using GridION (Oxford Nanopore Technology), using V.3 of the ARTIC protocol[13] and bioinformatics pipeline.[14] Samples were selected for sequencing if the corrected CT value was 33 or below, or the Hologic Aptima assay was above 1000 Relative Light Units (RLU). During the first wave, sequencing occurred between 1–31 March, while sequencing in the second wave restarted in November 2020–March 2021. Variants were called using updated versions of pangolin V.2.0.[15] We considered all cases in wave 1 to be non-Alpha variants, as our wave 1 cut-off of 25 July 2020 was 6 weeks prior to first identified cases of the Alpha variant in the UK[16] and before the Alpha variant was first identified in our population in November 2020.

### Data sources, extraction and integration

Clinical, laboratory and demographic data for all cases with a laboratory-reported SARS-CoV-2 PCR RNA test

on nose and throat swabs or lower respiratory tract specimens were extracted from hospital electronic health record data sources using records closest to the test date. Data were linked to the Index of Multiple Deprivation. Age, sex and ethnicity were extracted from the Electronic Patient Record (EPR). Self-reported Office for National Statistics (ONS) ethnic categories were stratified into white (British, Irish, Gypsy and white–other) or non-white (black (African, Caribbean and black–other) or Asian (Bangladeshi, Chinese, Indian, Pakistan and Asian–other) and mixed/other). Numbers for which data were missing are listed by each variable. Comorbidities and medication history were extracted from the EPR and e-noting using natural language processing (NLP). If a comorbidity was not recorded, it was assumed not to be present. Cases were characterised as having/not having a medical history of hypertension, cardiovascular disease (stroke, transient ischaemic attack, atrial fibrillation, congestive heart failure, ischaemic heart disease, peripheral artery disease or atherosclerotic disease), diabetes mellitus, chronic kidney disease, chronic respiratory disease (chronic obstructive pulmonary disease, asthma, bronchiectasis or pulmonary fibrosis) and neoplastic disease (solid tumours, haematological neoplasias or metastatic disease). Obesity was defined as either obesity present in the notes or recorded body mass index $\geq 30 \, kg/m^2$. Medicines data were extracted using both structured queries and NLP tools with medical and drug dictionaries. Additionally, checks on free text data were performed by a cardiovascular clinician to ensure the information was accurate.

Analysis was carried out on the secure Rosalind high-performance computer infrastructure[17] running Jupyter Notebook V.6.0.3, R V.3.6.3 and Python V.3.7.6.

### Statistical analysis and outcome measures

Descriptive statistics were summarised with mean and standard deviation for continuous variables if the distribution is normal, and the median and IQR if the distribution are non-normal. Count and percentages were used for categorical variables. For the comparisons of variables for wave 1 versus wave 2 variables, Alpha variant versus non-Alpha variants, as well as sequenced patients versus non-sequenced patients in wave 2, Kruskal-Walllis test was used for continuous variables and $\chi^2$ test for categorical variables with significance level of $p<0.05$. Multivariate analysis was performed using logistic regression to assess the odds ratios of different risk factors (including age, sex, ethnicity (white, non-white and unknown), variant status (Alpha or non-Alpha), and cardiovascular disease, hypertension, diabetes, chronic respiratory disease, cancer, kidney disease, HIV, transplant and frailty) for hypoxia on admission as the binary outcome indicating severity at admission. Variables to be included in the multivariate analysis were chosen

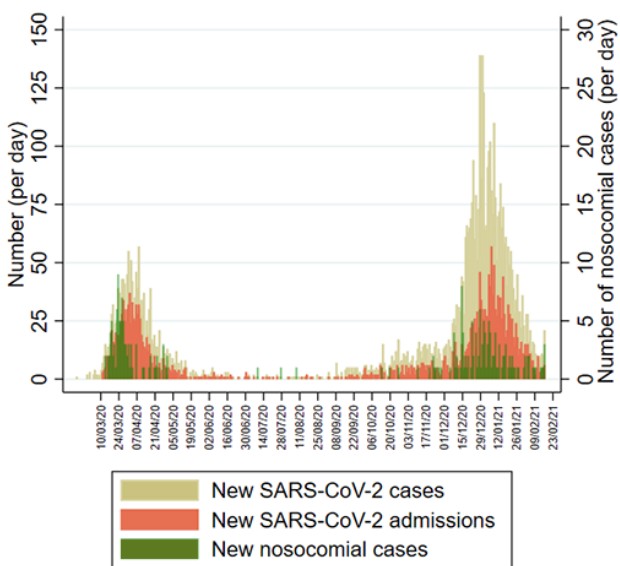

**Figure 1** Distribution of laboratory-confirmed SARS-CoV-2 cases over time. Daily incidence of new cases (beige), newly admitted cases (orange) and nosocomial acquisitions (green) over time.

by literature review and expert opinion (see online supplemental material). Cases with missing data points were dropped from analysis.

### RESULTS

#### General epidemiology and results of viral genome sequencing

Figure 1 shows the incidence of SARS-CoV-2 cases, SARS-CoV-2 admissions and nosocomial cases since 13 March 2020. In total, 5810 individuals had a positive SARS-CoV-2 PCR test up until the data extraction date of 17 February 2021. Two 'waves' are evident with 25 July taken as a separation date between waves, at which point a minimum of 12 wave 1 cases remained in the hospital. Wave 1 comprised 1528 cases (26.3%) from when laboratory testing commenced on 13 March to peak rapidly between 1 and 8 April 2020 with 57 new cases per day, before falling to a baseline by 12 May 2020. Ninety-one per cent (1391/1528) of all cases in wave 1 occurred during these 60 days. Wave 2 comprised 4282 cases (73.7%), with incidence first increasing gradually from the beginning of October. There was then a period of rapidly escalating incidence from about 10 December, peaking on 28 December 2020 when 139 cases per day were diagnosed. Of 4282 wave 2 cases, 3446 (80%) were detected during a comparable 60-day period between 10 December 2020 and 8 February 2021. In both waves, nosocomial cases peaked early, increasing along with admissions but then fell while the number of community admissions continued at peak levels.

Individuals with a positive test were placed into six categories (figure 2). The 5810 SARS-CoV-2 cases were categorised as follows: inpatients admitted within 14 days of a positive test (n=2341), HCWs (n=1549), outpatients

A

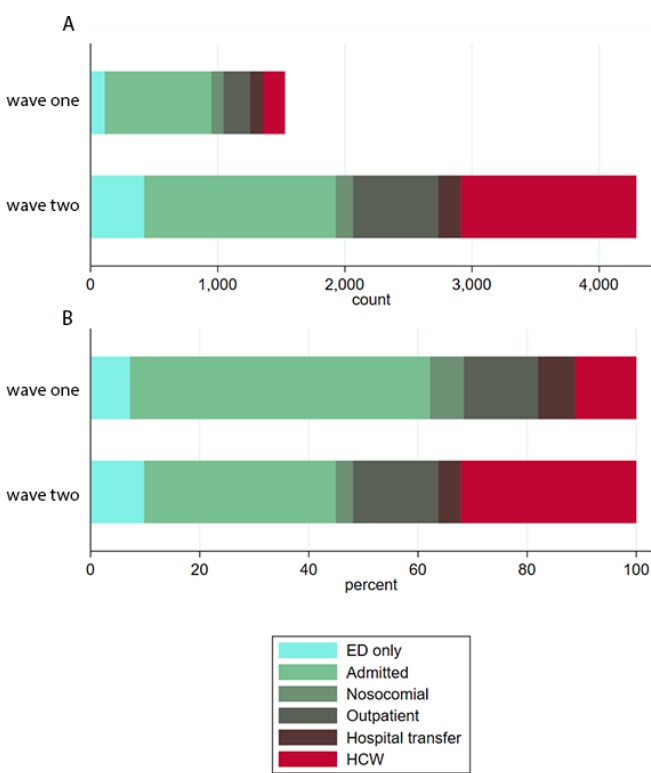

B

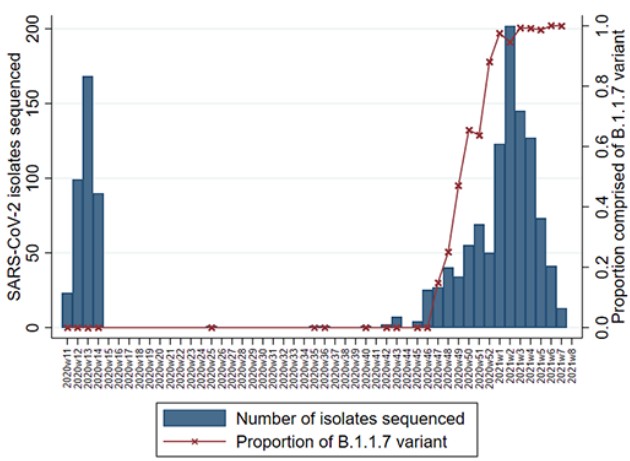

**Figure 3** Number of cases with sequenced SARS-CoV-2 isolates by epi-week (bar) and the proportion of which were made up of the Alpha variant B.1.1.7 (red line).

**Figure 2** (A) Absolute number of cases within the different hospital cohorts during wave 1 (upper) and wave 2 (lower). (B) Proportion of cases within the different hospital cohorts during wave 1 (upper) and wave 2 (lower). ED, emergency department; HCW, healthcare worker.

(n=874), ED attenders not subsequently admitted (n=532), interhospital transfers (n=281) and nosocomial cases (n=233). Some observed differences between waves 1 and 2 reflected the increased availability of testing particularly for outpatients (208, 13.6%, vs 666, 15.6%), people sent home from ED (111, 7.3%, vs 421, 9.8%) and HCWs (171, 11.2%, vs 1378, 32.2%). There were also more interhospital transfers of known COVID-19 cases in wave 2 (177, 4.1%, vs 104, 6.8%, in wave 1). In wave 2, the number of admissions increased (1503, 35.1%, vs 838, 54.8%) along with nosocomial cases (137, 3.2%, vs 96, 6.3%) compared with wave 1.

Figure 3 shows the 1470 successfully sequenced SARS-CoV-2 isolates over time, with 382 from wave 1 and 1088 from wave 2. Sequencing was successful for 216/838 (26%) admitted cases from wave 1, 472/1503 (31%) admitted cases in wave 2, and 121/233 (52%) nosocomial cases. The proportion of Alpha variant increased rapidly after the first Alpha isolate was identified on 15 November 2020, accounting for approximately two-thirds within 3 weeks, and almost 100% (600/617 isolates, 97%) in January 2021. In the second wave, the Alpha variant made up 83% (908/1088) of all sequenced isolates, 85% (400/472) of sequenced isolates from admitted cases and 88% (51/59) of sequenced isolates from nosocomial cases. In addition, two cases of the B.1.351 beta variant

of concern were also detected in the wave 2 admission cohort.

## Comparison of characteristics of admitted cases between waves 1 and 2

Descriptive statistics of cases admitted during wave 1 (n=838) and wave 2 (n=1503) were compared (table 1). There was a statistically significant difference in median age of 2 years (62 years in wave 1 vs 60 years in wave 2, p=0.019), and admitted cases were more likely to be female in wave 2 (47.3% vs 41.8%, p=0.011). A larger proportion of admitted cases in wave 2 were obese (29.1% vs 24.6%, p=0.02). Comparison of comorbidities showed that those in wave 2 were less likely to have a diagnosis of frailty (11.5% vs 22.8%, p<0.001), history of stroke (4.3% vs 8.6%, p<0.001) or cancer (4.8% vs 7.2%, p=0.022). There was no significant difference in proportion with known comorbidities of diabetes, kidney disease, hypertension, cardiovascular disease or respiratory disease.

There were no significant differences between waves in the proportion with severe SARS-CoV-2 disease on admission as judged by hypoxia (64.3% in wave 1 vs 65.5% in wave 2, p=0.67) or tachypnoea (respiratory rate >20 breaths/min) (23.9% vs 24.3%, p=0.86). There were small differences in other physiological parameters on admission, some of which reached statistical significance, but differences were not clinically relevant.

Laboratory markers were compared between waves (table 1). There were small but significant differences, such as lower C reactive protein (CRP) (median 51.0 mg/dL, IQR 18.0–103.8, vs 74.5 mg/dL, IQR 26.0–148.0; p<0.001) and lower ferritin (699.0, IQR 342.0–1359.0, vs 855.0, IQR 394.0–1533.5; p=0.05) in wave 2. There were other small statistically significant differences without clear clinical significance, such as a lower D-dimer in wave 2 (0.9 mg/L fibrinogen equivalent units (FEU), IQR 0.5–2.2, vs 1.1 mg/L FEU, IQR 0.6–3.0; p=0.001) and lower estimated glomerular filtration rate (69.0 mL/min,

**Table 1** Descriptive statistics of the cohort for wave 1 (n=838) and wave 2 (n=1503) admissions

| | Missing | Wave 1 n (%) | Wave 2 n (%) | Wave 1 Median (IQR) | Wave 2 Median (IQR) | P value |
|---|---|---|---|---|---|---|
| **Demographics** | | | | | | |
| Age (years) | 0 | | | 62.0 (49.0–78.0) | 60.0 (47.0–74.0) | 0.019 |
| Male | 0 | 488 (58.2) | 792 (52.7) | | | 0.011 |
| Ethnicity | 0 | | | | | 0.013 |
| White | | 331 (39.5) | 598 (39.8) | | | |
| Asian | | 64 (7.6) | 121 (8.1) | | | |
| Black–African | | 177 (21.1) | 262 (17.4) | | | |
| Black–Caribbean | | 73 (8.7) | 98 (6.5) | | | |
| Mixed | | 15 (1.8) | 18 (1.2) | | | |
| Other | | 45 (5.4) | 107 (7.1) | | | |
| Unknown | | 133 (15.9) | 299 (19.9) | | | |
| BMI | 577 | | | 27.0 (23.8–31.7) | 27.7 (24.0–32.9) | 0.022 |
| >30 | | 206 (24.6) | 438 (29.1) | | | 0.02 |
| >40 | | 34 (4.1) | 86 (5.7) | | | 0.098 |
| **Physiological parameters** | | | | | | |
| Heart rate (beats/min) | 360 | | | 84.0 (75.0–94.0) | 81.0 (72.0–91.0) | <0.001 |
| >100 | | 105 (12.5) | 142 (9.4) | | | 0.02 |
| Blood pressure (mm Hg) | | | | | | |
| Systolic | 369 | | | 125.0 (113.0–139.0) | 127.0 (115.0–141.0) | 0.013 |
| Diastolic | 369 | | | 73.0 (65.0–80.0) | 75.0 (68.0–82.0) | <0.001 |
| MAP | 369 | | | 90.7 (82.2–99.0) | 92.3 (84.7–101.3) | <0.001 |
| Respiratory rate (breaths/min) | 359 | | | 19.0 (18.0,22.0) | 19.0 (18.0–22.0) | 0.764 |
| >20 | | 200 (23.9) | 365 (24.3) | | | 0.86 |
| Hypoxia | 658 | 370 (64.3) | 726 (65.5) | | | 0.67 |
| Temperature (°C) | 361 | | | 36.9 (36.4–37.5) | 36.6 (36.2–37.2) | <0.001 |
| NEWS2 | 405 | | | | | 0.86 |
| 0 | | 95 (11.3) | 173 (11.5) | | | |
| 1 | | 108 (12.9) | 192 (12.8) | | | |
| 2 | | 117 (14.0) | 188 (12.5) | | | |
| >2 | | 371 (44.3) | 692 (46.0) | | | |
| **Laboratory parameters** | | | | | | |
| Neutrophils (×10$^9$/L) | 8 | | | 4.9 (3.4–7.6) | 5.0 (3.3–7.5) | 0.724 |
| Lymphocytes (×10$^9$/L) | 7 | | | 0.9 (0.6–1.3) | 0.9 (0.6–1.4) | 0.741 |
| NLR | 8 | | | 5.4 (3.1–9.9) | 5.4 (3.2–9.8) | 0.951 |
| Creatinine (µmol/L) | 43 | | | 83.0 (64.0–115.0) | 86.0 (68.0–117.0) | 0.065 |
| Urea (mmol/L) | 855 | | | 7.0 (4.6–12.2) | 6.0 (4.3–9.9) | 0.001 |
| Estimated GFR (mL/min) | 114 | | | 73.0 (48.0–98.0) | 69.0 (48.0–89.0) | 0.001 |
| Albumin (g/L) | 185 | | | 37.0 (32.0–40.0) | 38.0 (34.0–41.0) | <0.001 |
| CRP (mg/L) | 61 | | | 74.5 (26.0–148.0) | 51.0 (18.0–103.8) | <0.001 |
| D-dimer (mg/L FEU) | 1297 | | | 1.1 (0.6–3.0) | 0.9 (0.5–2.2) | 0.001 |
| Ferritin (µg/L) | 905 | | | 855.0 (394.0–1533.5) | 699.0 (342.0–1359.0) | 0.05 |
| **Comorbidities** | | | | | | |
| Stroke | 0 | 72 (8.6) | 64 (4.3) | | | <0.001 |

Continued

| | Missing | Wave 1 n (%) | Wave 2 n (%) | Wave 1 Median (IQR) | Wave 2 Median (IQR) | P value |
|---|---|---|---|---|---|---|
| **Table 1** Continued | | | | | | |
| TIA | 0 | 9 (1.1) | 20 (1.3) | | | 0.731 |
| Hypertension | 0 | 288 (34.4) | 464 (30.9) | | | 0.091 |
| Diabetes | 0 | 246 (29.4) | 384 (25.5) | | | 0.052 |
| AF | 0 | 63 (7.5) | 115 (7.7) | | | 0.972 |
| IHD | 0 | 146 (17.4) | 244 (16.2) | | | 0.495 |
| Heart failure | 0 | 54 (6.4) | 105 (7.0) | | | 0.679 |
| COPD | 0 | 64 (7.6) | 109 (7.3) | | | 0.796 |
| Asthma | 0 | 74 (8.8) | 138 (9.2) | | | 0.835 |
| Cancer | 0 | 60 (7.2) | 72 (4.8) | | | 0.022 |
| Kidney disease | 0 | 112 (13.4) | 181 (12.0) | | | 0.389 |
| HIV | 0 | 21 (2.5) | 36 (2.4) | | | 0.979 |
| Solid organ transplant | 0 | 24 (2.9) | 49 (3.3) | | | 0.686 |
| Frailty | 0 | 191 (22.8) | 173 (11.5) | | | <0.001 |

P value was from Kruskal-Wallis test for continuous variables and $\chi^2$ test for categorical variables.

AF, atrial fibrillation; BMI, body mass index; COPD, chronic obstructive pulmonary disease; CRP, C reactive protein; FEU, fibrinogen equivalent units; GFR, glomerular filtration rate; IHD, ischaemic heart disease; MAP, mean arterial pressure; NEWS2, National Early Warning Score 2; NLR, neutrophil and lymphocyte ratio; TIA, transient ischaemic attack.

IQR 48.0–89.0, vs 73.0 mL/min, IQR 48.0–98.0; p=0.001), lower urea (6.0 mmol/L, IQR 4.3–9.3, vs 7.0 mmol/L, IQR 4.6–12.2; p=0.001) and higher albumin (38.0 g/L, IQR 34.0–41.0 g/L, vs 37.0 g/L, IQR 32.0–40.0; p<0.001). There was no significant difference with neutrophils, lymphocytes, neutrophil and lymphocyte ratio, creatinine, and glucose.

### Comparison of characteristics of admitted cases infected with Alpha and non-Alpha variants

Given the reported association between increased disease severity and transmission with the Alpha variant, we compared demographic, physiological and laboratory parameters between admitted cases with infection caused by Alpha variant (n=400) with non-Alpha (n=910) variants (table 2).

Groups with non-Alpha and Alpha variants were not significantly different in median age (62 years vs 64 years, p=0.22) or ethnicity. The proportion of admissions who were female was larger in the group infected with the Alpha variant compared with those infected by non-Alpha variants (48.0% vs 41.8%, p=0.01).

Cases infected with the Alpha variant were less likely to be frail (14.5% vs 22.4%, p=0.001). A higher proportion of those in the Alpha variant group were obese (30.2% v 24.8%, p=0.048). Other minor differences in comorbidities between groups are shown in table 2 but did not reach statistical significance.

On admission, a higher proportion of those infected with the Alpha variant were hypoxic (70.0% vs 62.5%, p=0.029), the main indicator of severe disease. CRP on admission was lower in the Alpha variant group (54 mg/L, IQR 24.0–102.0) compared with those infected with non-Alpha variants (70 mg/L, IQR 25.0–142.0; p<0.001). Differences in other laboratory parameters did not meet either statistical or clinical significance.

### Multivariate analysis of factors associated with severity of COVID-19 on admission

Multivariate logistic regression was applied to look at associations with severity of disease on admission as measured by hypoxia (table 3), equivalent to WHO ordinal scale of ≥4.[12] Age, sex, ethnicity, comorbidities and variant status (Alpha vs non-Alpha) were entered into the model. Severity of disease on admission, as measured by hypoxia, was the outcome variable. Age was a significant predictor of severity, with an OR of 1.02 (95% CI 1.01 to 1.03, p<0.001) for hypoxia on admission for every advancing year. Obesity was associated with severity, giving an OR of 1.70 (95% CI 1.28 to 2.26, p<0.001). Infection with the Alpha variant was also associated with increased hypoxia on admission (OR 1.68, 95% CI 1.26 to 2.24; p<0.001). Other variables were not significantly associated with hypoxia on admission, including sex, ethnicity and comorbidities.

### Comparison of non-sequenced and sequenced cases in wave 2

We assessed for differences between the non-sequenced and sequenced inpatient cases to identify any possible bias in those that were sequenced. Demographics, admission physiological and laboratory parameters, and the outcome measure of hypoxia on admission are presented in table 4. There was no significant difference of the proportion with the outcome measure, hypoxia on admission, in both the sequenced and non-sequenced

**Table 2** Descriptive statistics of the cohort for non-Alpha variant (n=910) and Alpha variant (n-400) admissions

| | Missing | Non-Alpha variant n (%) | Alpha variant n (%) | Non-Alpha variant value (IQR) | Alpha variant value (IQR) | P value |
|---|---|---|---|---|---|---|
| **Demographics** | | | | | | |
| Age (years) | 0 | | | 62.0 (49.0–78.0) | 64.0 (52.0–78.0) | 0.22 |
| Male | | 530 (58.2) | 208 (52.0) | | | 0.042 |
| Ethnicity | 0 | | | | | 0.402 |
| White | | 358 (39.3) | 164 (41.0) | | | |
| Asian | | 71 (7.8) | 38 (9.5) | | | |
| Black–African | | 191 (21.0) | 67 (16.8) | | | |
| Black–Caribbean | | 78 (8.6) | 27 (6.8) | | | |
| Mixed | | 16 (1.8) | 6 (1.5) | | | |
| Other | | 50 (5.5) | 23 (5.8) | | | |
| Unknown | | 146 (16.0) | 75 (18.8) | | | |
| BMI | 334 | | | 27.1 (23.8–31.7) | 28.1 (24.0–34.2) | 0.036 |
| >30 | | 226 (24.8) | 121 (30.2) | | | 0.048 |
| >40 | | 36 (4.0) | 26 (6.5) | | | 0.063 |
| **Physiological parameters** | | | | | | |
| Heart rate (beats/min) | 198 | | | 84.0 (74.0–94.0) | 80.0 (72.0–90.0) | 0.001 |
| >100 | | 118 (13.0) | 36 (9.0) | | | 0.05 |
| Blood pressure (mm Hg) | | | | | | |
| Systolic | 201 | | | 125.0 (113.0–139.5) | 127.0 (115.0–142.0) | 0.138 |
| Diastolic | 201 | | | 73.0 (65.0–80.0) | 75.0 (67.0–83.0) | 0.01 |
| MAP | 201 | | | 90.7 (82.3–99.2) | 92.7 (84.0–101.7) | 0.022 |
| Respiratory rate (breaths/min) | 194 | | | 19.0 (18.0–21.0) | 19.0 (18.0–22.0) | 0.591 |
| >20 | | 209 (23.0) | 96 (24.0) | | | 0.737 |
| Hypoxia | 0 | 392 (62.5) | 217 (70.0) | | | 0.029 |
| Temperature (°C) | 199 | | | 36.9 (36.4–37.5) | 36.6 (36.2–37.1) | <0.001 |
| NEWS2 | 218 | | | | | 0.038 |
| 0 | | 107 (11.8) | 43 (10.8) | | | |
| 1 | | 125 (13.7) | 39 (9.8) | | | |
| 2 | | 127 (14.0) | 53 (13.2) | | | |
| >2 | | 391 (43.0) | 207 (51.7) | | | |
| **Laboratory parameters** | | | | | | |
| Neutrophils (×10$^9$/L) | 2 | | | 4.9 (3.4–7.6) | 4.8 (3.3–6.9) | 0.479 |
| Lymphocytes (×10$^9$/L) | 1 | | | 0.9 (0.6–1.3) | 0.8 (0.5–1.2) | 0.005 |
| NLR | 2 | | | 5.4 (3.1–9.9) | 5.8 (3.5–10.2) | 0.195 |
| Creatinine (µmol/L) | 16 | | | 83.0 (64.0–115.0) | 92.0 (74.0–126.0) | <0.001 |
| Urea (mmol/L) | 536 | | | 6.8 (4.3–12.0) | 6.6 (4.4–10.6) | 0.573 |
| Estimated GFR (mL/min) | 43 | | | 73.0 (48.5–97.0) | 63.5 (44.0–81.0) | <0.001 |
| Albumin (g/L) | 107 | | | 37.0 (33.0–41.0) | 38.0 (34.0–41.0) | 0.009 |
| CRP (mg/L) | 21 | | | 70.0 (25.0–142.0) | 54.0 (24.0–102.0) | <0.001 |
| D-dimer (mg/L FEU) | 727 | | | 1.1 (0.6–2 .8) | 0.9 (0.5–1.9) | 0.019 |
| Ferritin (µg/L) | 501 | | | 815.0 (366.2–1499.0) | 712.0 (357.5–1294.0) | 0.341 |
| **Comorbidities** | | | | | | |
| Stroke | 0 | 74 (8.1) | 22 (5.5) | | | 0.117 |
| TIA | 0 | 12 (1.3) | 5 (1.2) | | | 0.87 |
| Hypertension | 0 | 315 (34.6) | 144 (36.0) | | | 0.674 |

Continued

**Table 2**    Continued

| | Missing | Non-Alpha variant n (%) | Alpha variant n (%) | Non-Alpha variant value (IQR) | Alpha variant value (IQR) | P value |
|---|---|---|---|---|---|---|
| Diabetes | 0 | 267 (29.3) | 106 (26.5) | | | 0.326 |
| AF | 0 | 72 (7.9) | 42 (10.5) | | | 0.154 |
| IHD | 0 | 162 (17.8) | 78 (19.5) | | | 0.513 |
| Heart failure | 0 | 61 (6.7) | 34 (8.5) | | | 0.299 |
| COPD | 0 | 74 (8.1) | 32 (8.0) | | | 0.977 |
| Asthma | 0 | 84 (9.2) | 39 (9.8) | | | 0.846 |
| Cancer | 0 | 64 (7.0) | 21 (5.2) | | | 0.278 |
| Kidney disease | 0 | 122 (13.4) | 62 (15.5) | | | 0.359 |
| HIV | 0 | 22 (2.4) | 10 (2.5) | | | 0.916 |
| Solid organ transplant | 0 | 25 (2.7) | 19 (4.8) | | | 0.092 |
| Frailty | 0 | 204 (22.4) | 58 (14.5) | | | 0.001 |

P value was from Kruskal-Wallis test for continuous variables and $\chi^2$ test for categorical variables.
AF, atrial fibrillation; BMI, body mass index; COPD, chronic obstructive pulmonary disease; CRP, C reactive protein; FEU, fibrinogen equivalent units; GFR, Glomerular Filtration Rate; IHD, ischaemic heart disease; MAP, mean arterial pressure; NEWS2, National Early Warning Score 2; NLR, neutrophil and lymphocyte ratio; TIA, transient ischaemic attack.

inpatient groups (47% vs 50%, p=0.381). There was no significant difference in the proportion of men in the sequenced group compared with the non-sequenced group (52.2% vs 53.8%, p=0.595) as with obesity (39.5% vs 38.4%, p=0.783) or the proportion of those from non-white ethnic backgrounds (41.4% vs 40.5%, p=0.934). On average, sequenced inpatient cases were significantly older (63 vs 57 years, p<0.001) and had a larger proportion of some comorbidities than non-sequenced cases.

**Table 3**    ORs for severity (hypoxia) at admission from multivariate logistic regression model

| | OR | P value | 95% CI |
|---|---|---|---|
| Age | 1.02 | <0.001 | 1.01 to 1.03 |
| Male | 0.96 | 0.75 | 0.73 to 1.25 |
| Ethnicity | | | |
| Non-white | 1.15 | 0.35 | 0.86 to 1.55 |
| Unknown | 1.20 | 0.36 | 0.81 to 1.77 |
| Comorbidity | | | |
| Body mass index >30 | 1.70 | <0.001 | 1.28 to 2.26 |
| Cardiovascular | 0.79 | 0.15 | 0.58 to 1.09 |
| Hypertension | 1.11 | 0.52 | 0.81 to 1.51 |
| Diabetes | 0.75 | 0.07 | 0.55 to 1.02 |
| Chronic respiratory disease | 1.20 | 0.32 | 0.83 to 1.74 |
| Cancer | 0.60 | 0.06 | 0.35 to 1.02 |
| Kidney disease | 0.74 | 0.17 | 0.48 to 1.14 |
| HIV | 1.74 | 0.16 | 0.80 to 3.78 |
| Organ transplant | 0.79 | 0.55 | 0.37 to 1.71 |
| Frailty | 0.96 | 0.85 | 0.64 to 1.45 |
| Alpha variant | 1.68 | <0.001 | 1.26 to 2.24 |

## DISCUSSION

Our data from a large, multisite healthcare institution in one of the worst affected regions internationally provide a large dataset for in-depth comparison; for instance, we report a similar number of cases as reported from a national observational cohort study from Japan.[18] Our hospitalised cohort shares similar demographics to other city populations in the UK, representative of London with around 40% of individuals from non-white ethnicities.[19] This compares to national population studies where the average age of cases was much lower and with lower proportion from non-white ethnicities.[8 20]

There were threefold more SARS-CoV-2 RNA positive cases reported by the hospital laboratory in wave 2. Partly, this is attributed to increased testing capacity and changing testing strategy throughout 2020 (online supplemental figure 1). Due to capacity limits, during wave 1, it was not local policy to offer testing to outpatients and those not requiring admission, instead relying on clinical diagnosis. HCWs were not offered occupational health testing until the end of wave 1. We therefore restricted comparison to inpatient and nosocomial cases.

There were almost twice as many admitted cases in wave 2 compared with wave 1 (1503 vs 838). This is consistent with a higher local community incidence as reported by the ONS infection survey with 3.5% of individuals in London infected in January 2021,[21] compared with 2.2% of individuals in London at the peak of wave 1.[1] The increase in peak hospital occupancy in wave 2 has also been reported nationally.[22] A major contributor to this increase in hospital admissions is likely to be the emergence of the Alpha variant, which is reported to be more transmissible.[7]

Our finding is the first study in hospitalised cohorts to show increased severity of disease with the Alpha variant,

**Table 4** Patient characteristics of sequenced and non-sequenced inpatients in wave 2

|  | Non-sequenced | Sequenced | P value |
|---|---|---|---|
| n | 1031 | 472 |  |
| Age (SD) | 57.3 (21.0) | 62.9 (19.9) | <0.001 |
| Male (%) | 538 (52.2) | 254 (53.8) | 0.595 |
| Ethnicity (%) |  |  | 0.934 |
| White | 418 (40.5) | 194 (41.1) |  |
| Non-white | 417 (40.4) | 192 (40.7) |  |
| Unknown | 196 (19.0) | 86 (18.2) |  |
| Comorbidities |  |  |  |
| Body mass index >30 (%) | 302 (38.4) | 139 (39.5) | 0.783 |
| Cardiovascular (%) | 218 (21.1) | 142 (30.1) | <0.001 |
| Hypertension (%) | 300 (29.1) | 172 (36.4) | 0.005 |
| Diabetes (%) | 269 (26.1) | 127 (26.9) | 0.787 |
| Chronic respiratory disease (%) | 143 (13.9) | 82 (17.4) | 0.091 |
| Cancer (%) | 46 (4.5) | 26 (5.5) | 0.452 |
| Kidney disease (%) | 116 (11.3) | 74 (15.7) | 0.021 |
| HIV (%) | 26 (2.5) | 11 (2.3) | 0.966 |
| Organ transplant (%) | 31 (3.0) | 18 (3.8) | 0.509 |
| Frailty (%) | 108 (10.5) | 76 (16.1) | 0.003 |
| Hypoxia (%) | 491 (47.6) | 237 (50.2) | 0.381 |

as defined by hypoxia on admission, which is equivalent to WHO ordinal scale of ≥4[12] and a key marker of severe disease. The validity of using hypoxia as a marker of severity is shown by the clinical characteristics of SARS-CoV-2, with respiratory illness causing hypoxia in a minority of cases and with a smaller proportion having respiratory failure necessitating ventilation.[23] Hypoxia on admission was chosen as a marker of severity to prevent confounding of results by changes in management of hospitalised patients across the pandemic. For instance steroid treatment, which was introduced during the study period around November 2020, have been shown to reduce risk of ventilation and death.[24] Other improvements in management, such as proning, anticoagulation and tocilizumab, could also confound other severity outcomes like death and intensive care unit (ICU) admission. Hypoxia on admission is not at risk of confounding by changes in management of cases, as currently no significant management or treatment options are deployed in the community.

Our finding of increased severity with the Alpha variant is consistent with that reported in community studies, which show increased hospitalisation[20] and mortality[8] with a similar hazard to which we find here for hypoxia on admission. Notably however, these community studies failed to control for comorbidities.[8 20] The association with severity we find persists even after adjustment for age, sex and comorbidities. Moreover, testing in the first wave prior to emergence of the Alpha variant was strict due to limited testing capacity, potentially leading to an ascertainment bias towards more severe cases in the first wave.

In comparison, in the second wave, testing was more widespread, potentially leading to increased ascertainment of less severe cases. This makes it even more striking that the association of the Alpha variant, which dominated the second wave, with severe disease is so prominent.

Notably, the only other published study in hospital cohorts showed no difference in severity as measured by the composite outcome of need for ventilation or death.[10] Broadly, the two cohorts from these hospital cohorts are similar, with an average age of around 60 and a high proportion of non-white ethnicities. In general, this supports the external validity of our findings, but replication in dissimilar cohorts are awaited. The difference between findings in our study and those of Frampton et al[10] may be related to the choice of outcome. Our choice of outcome, hypoxia on admission, represents the natural history of disease prior to medical intervention as no treatments are currently deployed in the community. The mortality outcome investigated by Frampton et al is after hospital treatment, which may ameliorate the severity increase that we find with the Alpha variant, thereby explaining the differences in severity seen between our studies. Interestingly, despite male sex being widely reported to be a risk factor for severe disease, our multivariate model confirms findings by these authors that sex is not significantly associated with severity in hospitalised cohorts after adjusted analysis.[10]

The lack of association between severity and male sex may correspond to the increase in the proportion of women in the admitted cohort of wave 2 and those infected with Alpha, accounting for an extra 5% of

admissions with SARS-CoV-2 infection. A study in press[25] suggests the Alpha variant may be more severe in hospitalised women who may have increased mortality and/or requirement for ICU care. Our data, showing an increase in the proportion of women in the admission cohort and lack of expected association of severity with male sex is consistent with the finding that Alpha may show increased virulence in women.

We also included an assessment of bias by comparing characteristics of non-sequenced cases with those successfully sequenced. While sequenced patients were older and more comorbid, there was no significant difference between the proportion with the outcome measure of hypoxia on admission between our sequenced and non-sequenced cases. This suggests no significant bias towards severity in the sequenced group, which was predominantly made up of cases of the Alpha variant.

Admitted cases in wave 2 were also around half as likely to have a diagnosis of frailty, which may be due to fewer admissions from care homes during wave 2, which has been reported both nationally[26] and internationally.[27] Additionally, admitted cases were around a third less likely to have cancer in wave 2. Both of these reductions may also be as a result of individuals shielding, and therefore at reduced risk of acquiring SARS-CoV-2 infection. Other differences in comorbidities between waves were small and of unclear clinical significance.

One additional striking observation was the similarity in the number of nosocomial cases in wave 1 (n=96 of 934 (10%) inpatient cases) and wave 2 (n=137 of 1640 (8%) inpatient cases). This incidence of nosocomial infection is a major challenge for UK healthcare institutions, with associated crude mortality at around 30% during the first wave.[28 29] Interestingly, nosocomial cases in wave 1 increased and started to fall before impact of the main infection control interventions of banning hospital visitors (25 March), introducing universal surgical mask wearing (28 March 2020) and universal regular inpatient screening (after the first wave). In comparison, all these measures were in place prior to the second wave. The similar number of cases in wave 2 may in part be due to increased inpatient screening, which would identify asymptomatic cases, or introduction of the more transmissible Alpha variant, which made up the vast majority of our sequenced nosocomial cases.

Some healthcare institutions report far fewer nosocomial acquisitions; for instance, an academic hospital in Boston, USA, reported only two nosocomial cases in over 9000 admissions.[30] This could be due to greater availability of side rooms for isolation or their use of N95 masks by HCWs, which may decrease transmission between HCWs and patients. In contrast, current UK public health policy recommends surgical facemasks for patient interactions unless performing aerosol-generating procedures.[31] For this reason, it will be important to further investigate the factors involved in nosocomial acquisition in both waves.

One limitation of our study is that the population comes from one city, and findings therefore need to be compared with findings in other regions. Our dataset included cases confirmed by SARS-CoV-2 RNA testing in our laboratory and so may miss those diagnosed only clinically. We could not compare outcomes after hospital admission, such as ICU admission or mortality, due to changes in in-hospital management between waves. In addition, we were unable to include some variables associated with severity in other studies due to few cases with these features (eg, pregnancy) or due to poor coding in the dataset (eg, liver disease), which prevents us from commenting on the risk associated with these variables.

The number of cases diagnosed, admissions and nosocomial cases were higher in wave 2 than in wave 1, likely due to the increased incidence caused by the more transmissible Alpha variant. Infection with the Alpha variant was associated with severity as measured by hypoxia on admission, the first such finding in hospitalised cohorts. Our findings support growing evidence that emerging variants may have altered virulence as well as increased transmissibility, with such evidence providing support for public health efforts to contain their spread. More broadly, it also increases understanding of the emergence of novel pathogens as they adapt to human hosts.

**Author affiliations**
[1]Centre for Clinical Infection & Diagnostics Research, King's College London, London, UK
[2]Department of Infection, Guy's and St Thomas' NHS Foundation Trust, London, UK
[3]Department of Population Health Sciences, King's College London, London, UK
[4]Infection Sciences, Viapath, London, UK
[5]NIHR Biomedical Research Centre, Guy's and St. Thomas' NHS Foundation Trust, London, UK

**Acknowledgements** The authors acknowledge use of the research computing facility at King's College London, Rosalind (https://rosalind.kcl.ac.uk), which is delivered in partnership with the National Institute for Health Research (NIHR) Biomedical Research Centres at South London & Maudsley and Guy's and St Thomas' NHS Foundation Trusts, and partly funded by capital equipment grants from the Maudsley Charity (award 980) and Guy's and St Thomas' Charity (TR130505). The views expressed are those of the authors and not necessarily those of the NHS, the NIHR, King's College London or the Department of Health and Social Care.

**Collaborators** COG-UK HOCI Investigators. Barts site: Teresa Cutino-Moguel Barts Heath NHS Trust PI Barts Health, Tabassum Khan (Barts Heath NHS Trust, Research assistant), Beatrix Kele (Barts Heath NHS Trust, Sequencing scientist), Raghavendran Kulasegaran-Shylini (Barts Heath NHS Trust, Sequencing scientist), Claire E. Broad (Barts Heath NHS Trust, Sequencing scientist), Dola Owoyemi (Barts Heath NHS Trust, Sequencing scientist), David Harrington (Barts Heath NHS Trust, Infection Control Doctor), Clare Coffey (Barts Heath NHS Trust, Infection Control nurse), Martina Cummins (Barts Heath NHS Trust, Infection Control nurse), Anna Riddell (Barts Heath NHS Trust, Virology Consultant), Tyrra D'Souza (Barts Heath NHS Trust, Research Assistant). Glasgow site: Guy Mollett MRC-University of Glasgow Centre for Virus Research Clinical Research Fellow, Emma Thomson MRC-University of Glasgow Centre for Virus Research and NHS, Greater Glasgow and Clyde Principal Investigator Christine Peters, (NHS Greater Glasgow and Clyde Microbiology Consultant), Aleks Marek (NHS Greater Glasgow and Clyde Infection Control Lead/Microbiology Consultant), Rory Gunson (NHS Greater Glasgow and Clyde Virology laboratory lead), Emily Goldstein (NHS Greater Glasgow and Clyde Sample extraction), Emilie Shepherd (NHS Greater Glasgow and Clyde Sample extraction), James Shepherd (MRC-University of Glasgow Centre for Virus Research Clinical Research Fellow), David Robertson (MRC-University of Glasgow Centre for Virus Research Lead bioinformatician), Katherine Smollett (MRC-University of Glasgow Centre for Virus Research Sequencing), Ana da Silva Filipe (MRC-University of Glasgow Centre for Virus Research Sequencing), Alice Broos (MRC-University of Glasgow Centre for Virus Research Sequencing), Stephen Carmichael (MRC-

University of Glasgow Centre for Virus Research Sequencing), Nicholas Suarez (MRC-University of Glasgow Centre for Virus Research Sequencing), Chris Davis (MRC-University of Glasgow Centre for Virus Research Sample extraction), Sreenu Vattipally (MRC-University of Glasgow Centre for Virus Research Bioinformatician), Joseph Hughes (MRC-University of Glasgow Centre for Virus Research Bioinformatician), Ioulia Tsatsani (MRC-University of Glasgow Centre for Virus Research Bioinformatician) Jacqueline McTaggart (NHS Greater Glasgow and Clyde Research Nurse), Stephanie McEnhill (NHS Greater Glasgow and Clyde Research Nurse) Guy's and St Thomas' site: Adela Medina (Viapath, Sequence), Themoula Charalampous (KCL, Sequence), Bindi Patel (GSTT NHS Trust, Sequence), Flavia Flaviani (GSTT NHS Trust, Bioinformatics), Jörg SaBmannshausen (GSTT NHS Trust, Bioinformatics/IT), May Rabuya (GSTT NHS Trust Research, Nurse-data collection), Sulekha Gurung (GSTT NHS Trust Research, Nurse-data collection), Anu Augustine (GSTT NHS Trust Research, Nurse-data collection), Rahul Batra (GSTT NHS Trust Sequencing/IT/manager), Luke Snell (GSTT NHS Trust Sequence, bioinf, data collection, IPC), Gaia Nebbia, (GSTT NHS Trust Principal Investigator). Imperial site: Alison Holmes (Imperial Healthcare NHS Trust, Principal Investigator), Sid Mookerjee (Imperial Healthcare NHS Trust, Data lead), James Price (Imperial Healthcare NHS Trust Site IPC Lead), Paul Randell (Imperial Healthcare NHS Trust Laboratory Lead), Krystal Johnson (Imperial Healthcare NHS Trust Research Nurse),Thilipan Thaventhiran (Imperial Healthcare NHS Trust Research Nurse), Damien Mine (Imperial Healthcare NHS Trust Clinician), Sophie Hunter (Imperial Healthcare NHS Trust Research Nurse), Isa Ahmad (Imperial Healthcare NHS Trust, Data Analyst) Anitha Ramanathan (Imperial Healthcare NHS Trust Research Nurse). Liverpool site: Anu Chawla (Liverpool NHS Foundation Trust, Principal Investigator); Alistair Derby (University of Liverpool, Sequencing lab lead); Sam Haldenby (University of Liverpool, Bioinformatics lead); Becky Taylor (Liverpool NHS Foundation Trust; Research data coordinator); Keith Morris (Liverpool NHS Foundation Trust, Research nurse); Charles Numbere (Liverpool NHS Foundation Trust, Healthcare assistant); Mark Hopkins (Liverpool NHS Foundation Trust, Consultant clinical scientist); Jenifer Mason (Liverpool NHS Foundation Trust, Consultant microbiologist); Alexandra Bailey (Liverpool NHS Foundation Trust, Research administrator); Manchester site: Nicholas Machin (PHE and MFT, Principal Investigator); Shazaad Ahmad (MFT, Consultant Virologist and IPC Doctor: review of sequencing reports); Julie Cawthorne (MFT, Clinical Director of Infection Prevention and Control: review of sequencing reports and assistance with CRF completion); Ryan George (MFT, IPC surveillance officer: co-ordination of metadata and sequencing reports); James Montgomery (MFT, IPC Nurse: review of sequencing reports and implementation of IPC actions); Deborah McKew (MFT, IPC Nurse: review of sequencing reports and implementation of IPC actions); Newcastle site: Yusri Taha (Newcastle NHS Trust, Site PI); Angela Cobb (Newcastle NHS Trust; IPC matron); Michelle Ramsay (Newcastle NHS Trust, Infection Control); Maria Leader (Newcastle NHS Trust, Infection Control); Shirelle Burton-Fanning (Newcastle NHS Trust, Virologist); Julie Samuel (Newcastle NHS Trust, Microbiologist and IPC doctor); Sarah Francis (Newcastle NHS Trust, Trial coordinator); Lydia Taylor (Newcastle NHS Trust, Trial's Research Nurse); Darren Smith (Northumbria University, Lead, sequencing); Matthew Bashton (Northumbria University, Bioinformatics lead); Matthew Crown (Northumbria University, Bioinformatics scientist); Nottingham site: Nikunj Mahida (Nottingham NHS Trust, Principal Investigator); Matthew Loose (University of Nottingham, Sequencing/Bioinfomatics); Patrick McClure (University of Nottingham, Sequencing/Bioinfomatics); Mitch Clarke (Nottingham NHS Trust, IPC - IPC Lead - review of cases, sequencing data); Elaine Baxter (Nottingham NHS Trust, IPC - Senior IPC team member, review of cases, sequencing data); Carl Yates (Nottingham NHS Trust, IPC - Senior IPC team member, review of cases, sequencing data); Irfan Aslam (Nottingham NHS Trust, Data Entry); Vicki Fleming (Nottingham NHS Trust, Sample collection and processing); Michelle Lister (Nottingham NHS Trust, Sample collection and processing); Johnny Debebe (University of Nottingham, Bioinformatics); Nadine Holmes (University of Nottingham, Sequencing); Christopher Moore (University of Nottingham, Sequencing); Matt Carlile (University of Nottingham, Sequencing); Royal Free site: Tabitha Mahungu (Royal Free London NHS Trust, Principal Investigator); Sophie Weller (Royal Free London NHS Trust, Sub-Investigator); Tanzina Haque (Royal Free London NHS Trust, Sub-Investigator); Jennifer Hart (Royal Free London NHS Trust, Sub-Investigator); Dianne Irish-Tavares (Royal Free London NHS Trust, Sub-Investigator); Eric Witele (Royal Free London NHS Trust, Clinical Research Nurse); Mia De Mesa (Royal Free London NHS Trust, Clinical Research Nurse); Vicky Pang (Royal Free London NHS Trust, Head of Infection Prevention & Control Nursing – provided IPC data for CRFs); Jelena Heaphy (Royal Free London NHS Trust, Clinical Lead Nurse Infection Prevention and Control - provided IPC data for CRFs); Wendy Chatterton (Health Services Laboratory, Virology Service Manager, Organised samples & Logistics); Monika Pusok (Health Services Laboratory, Medical laboratory assistant, Organised samples & Logistics); Sandwell site: Tranprit Saluja (Sandwell & West Birmingham Hospitals NHS Trust, Principal Investigator - Consultant Microbiologist and IPC doctor); Zahira Maqsood (Sandwell NHS Trust, Clinical Research Practitioner); Angie Williams (Sandwell NHS Trust, Research Data Coordinators); Debbie Devonport (Sandwell NHS Trust, Research Data Coordinators); Lucy Palinkas (Sandwell NHS Trust, Infection control Data Analyst); Diane Thomlinson (Sandwell NHS Trust, Infection control Nurse); Julie Booth (Sandwell NHS Trust, Lead Nurse IPC); Ashok Dadrah (Sandwell NHS Trust, Laboratory Services Manager); Amanda Symonds (Sandwell NHS Trust, Senior Biomedical Scientist (Microbiology)); Cassandra Craig (Sandwell NHS Trust, Laboratory Associate Practitioner); Abhinav Kumar (Sandwell NHS Trust, Consultant microbiologist); Sheffield site: Thushan de Silva (University of Sheffield, Principal Investigator); Matthew D Parker (University of Sheffield, Bioinformatics processing/management); Peijun Zhang (University of Sheffield, WGS); Max Whiteley (University of Sheffield, WGS); Benjamin B Lindsey (University of Sheffield, WGS); Paige Wolverson (University of Sheffield, WGS); Benjamin H Foulkes (University of Sheffield, WGS); Luke Green (University of Sheffield, WGS); Marta Gallis Ramalho (University of Sheffield, WGS); Stavroula F Louka (University of Sheffield, WGS); Adrienn Angyal (University of Sheffield, WGS); Nikki Smith (University of Sheffield, Management/admin); David G Partridge (Sheffield NHS Trust, Investigator); Cariad Evans (Sheffield NHS Trust, Investigator); Mohammad Raza (Sheffield NHS Trust, Investigator); Hayley Colton (Sheffield NHS Trust, Investigator); Rebecca Gregory (Sheffield NHS Trust, Clinical trial assistant); Phillip Ravencroft (Sheffield NHS Trust, Clinical trial assistant); Katie Johnson (Sheffield NHS Trust, Sample collection and processing); Sharon Hsu (University of Sheffield, Bioinformatics support); Alexander J Keeley (Sheffield NHS Trust); Alison Cope (Sheffield NHS Trust); Amy State (Sheffield NHS Trust, Sample collection and processing); Nasar Ali (Sheffield NHS Trust); Rasha Raghei (Sheffield NHS Trust); Joe Heffer (Sheffield NHS Trust); Stella Christou (University of Sheffield, WGS); Samantha E Hansford (University of Sheffield, Management/admin); Hailey R Hornsby (University of Sheffield, WGS); Phil Wade (Sheffield NHS Trust, Data collection); Kay Cawthron (Sheffield NHS Trust, Data collection); Maqsood Khan (Sheffield NHS Trust, Data collection); Amber Ford (Sheffield NHS Trust, Data input); Imogen Wilson (Sheffield NHS Trust, Data input); Kate Harrington (Sheffield NHS Trust, Sample collection); Nic Tinker (Sheffield NHS Trust, Sample collection); Sally Nyinza (Sheffield NHS Trust, Investigator); Southampton site: Kordo Saeed (Southampton NHS Trust, Principal Investigator); Jacqui Prieto (Southampton NHS Trust, Samples/logistics); Adhyana Mahanama (Southampton NHS Trust, Samples/logistics); Buddhini Samaraweera (Southampton NHS Trust, Samples/logistics); Siona Silviera (Southampton NHS Trust, Samples/logistics); Emanuela Pelosi (Southampton NHS Trust, Samples/logistics); Eleri Wilson-Davies (Southampton NHS Trust, Samples/logistics); Sarah Jeremiah (Southampton NHS Trust, Data collection); Helen Wheeler (Southampton NHS Trust, Data collection); Matthew Harvey (Southampton NHS Trust, Data collection); Thea Sass (Southampton NHS Trust, Data collection); Helen Umpleby (Southampton NHS Trust, Data collection); Stephen Aplin (Southampton NHS Trust, Data collection); Samuel Robson (Portsmouth University, Sequencing lead); Sharon Glaysher (Portsmouth Hospital NHS Trust, Sequencing); Scott Elliott (Portsmouth Hospital NHS Trust, Sequencing); Kate Cook (Portsmouth University, Sequencing); Christopher Fearn (Portsmouth University, Sequencing); Salman Goudarzi (Portsmouth University, Sequencing); Katie Loveson (Portsmouth University, Sequencing); St Georges site: Kenneth Laing (St Georges, UoL, Sequencing); Irene Monahan (St Georges, UoL, Sequencing); Adam Witney (St Georges, UoL, Bioinformatician); Joshua Taylor (St Georges NHS Trust, Virology, data collection, CRF completion and upload to MACRO); NgeeKeong Tan (St Georges NHS Trust, Virology, data collection, CRF completion and upload to MACRO); Cassie Pope (St Georges NHS Trust and St Georges, UoL, PI, data collection, CRF completion and upload to Macro); Claudia Cardosa Pereira (St Georges NHS Trust, IPC nurse); Vaz Malik (St Georges, UoL, Upload to macro); UCLH site: Gee Yen Shin (UCLH NHS Trust, Principal Investigator, virologist); Eleni Nastouli (UCLH NHS Trust, Virologist); Catherine Houlihan (UCLH NHS Trust, Virologist); Judith Heaney (UCLH NHS Trust, Clinical scientist); Matt Byott (UCLH NHS Trust, Bioinformatician); Dan Frampton (UCL / UCLH, Bioinformatician); Gema Martinez-Garcia (UCLH NHS Trust, Senior infection control nurse); Leila Hail (UCLH NHS Trust, Senior infection control nurse); Ndifreke Atang (UCLH NHS Trust, Clinical trials practitioner); Helen Francis (UCLH NHS Trust, Research nurse); Milica Rajkov (UCLH NHS Trust, Clinical trials co-ordinator); UCL Genomics: Judith Breuer (UCL, Chief Investigator); Rachel Williams (UCL, Sequencing); Sunando Roy (UCL, Sequencing); Charlotte Williams (UCL, Sequencing); Nadua Bayzid (UCL, Sequencing); Marius Cotic (UCL, Sequencing); UCL Comprehensive Clinical Trials Unit: James Blackstone (UCL, Project Manager); Leanne Hockey (UCL, Trial Manager); Rachel McComish (UCL, Data Analyst); Alyson MacNeil (UCL, Trial Manager); Monica Panca (UCL, Health Economist); Georgia Marley (UCL, Data Manager); UCL Institute for Global Health: Andrew Copas (UCL, Senior Statistician); Oliver Stirrup (UCL, Statistician); Fiona Mapp (UCL, Qualitative Researcher); UCL Research IT Services; Alif Tamuri

(UCL, IT Developer); Stefan Piatek (UCL, IT Developer); University of Strathclyde: Paul Flowers (UoS, Senior Qualitative Researcher).

**Contributors** LBS and WW were involved in the conceptualisation, methodology, formal analysis of the synthesised data and writing (original drafting, review and editing). TC, AM and GN were involved in the investigation, being responsible for whole-genome sequencing and analysis of results. RB, FH and LdJ were involved in resources, administration and data curation. YW, JE and VC were involved in supervision, funding acquisition and drafting (review and editing). COG-UK HOCI provided funding. All authors agreed on the final manuscript. VC acts as guarantor to the study.

**Funding** FH, LBS, YW and VC are supported by the National Institute for Health Research (NIHR) Biomedical Research Centre programme of Infection and Immunity (RJ112/N027) based at Guy's and St Thomas' National Health Service NHS) Foundation Trust and King's College London. This work was also supported by The Health Foundation and the Guy's and St Thomas' Charity. COG-UK is supported by funding from the Medical Research Council part of UK Research & Innovation, the NIHR and Genome Research Limited, operating as the Wellcome Sanger Institute. VC is supported by the National Institute for Health Research (NIHR) Applied Research Collaboration South London (NIHR ARC South London) at King's College Hospital NHS Foundation Trust. The views expressed are those of the authors and not necessarily those of the NIHR or the Department of Health and Social Care.

**Competing interests** None declared.

**Patient consent for publication** Not applicable.

**Ethics approval** Ethical approval for data informatics was granted by The London Bromley Research Ethics Committee (reference (20/HRA/1871)) to the King's Health Partners Data Analytics and Modelling COVID-19 Group to collect clinically relevant data points from patients' electronic health records. Whole-genome sequencing of residual viral isolates was conducted under the COVID-19 Genomics UK (COG-UK) consortium study protocol, which was approved by the Public Health England Research Ethics and Governance Group (reference: R&D NR0195).

**Provenance and peer review** Not commissioned; externally peer reviewed.

**Data availability statement** Data are available in a public, open access repository. Sequencing data are available on Global Initiative on Sharing Avian Influenza Data (GISAID). Patient-level metadata are not otherwise available due to Research Ethics Committee (REC) approval.

**ORCID iDs**
Luke B Snell http://orcid.org/0000-0002-6263-9497
Wenjuan Wang http://orcid.org/0000-0002-1879-7332
Themoula Charalampous http://orcid.org/0000-0002-8914-5868
Gaia Nebbia http://orcid.org/0000-0002-7524-1947
Yanzhong Wang http://orcid.org/0000-0002-0768-1676

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
