## [Reviewer comments · BMJ Open]

ARTICLE DETAILS

TITLE (PROVISIONAL)	A descriptive comparison of admission characteristics between pandemic waves and multivariable analysis of the association of the Alpha variant (B.1.1.7 lineage) of SARS-CoV-2 with disease severity in inner London.
AUTHORS	Snell, Luke; Wang, Wenjuan; Medina, Adela; Charalampous, Themoula; Batra, Rahul; de Jongh, Leonardo; Higgins, Finola; Investigators, COG-UK HOCl; Nebbia, Gaia; Wang, Yanzhong; Edgeworth, Jonathan; Curcin, Vasa

VERSION 1 – REVIEW

REVIEWER	Ssematimba, Amos Gulu University, Department of Mathematics, Faculty of Science
REVIEW RETURNED	26-Jul-2021

GENERAL COMMENTS	Reviewer report Manuscript title: First and second SARS-CoV-2 waves in inner London: A comparison of admission characteristics and the association of the Alpha variant with disease severity Overall summary: The study addresses a key component on the yet-to-be understood aspects of COVID-19 namely, the impact of variants on the disease transmission dynamics. It uses field data that is suited for the analysis and among the method, they linked clinical data with viral genome sequence data to compare admitted cases between SARS-CoV-2 waves in London, and to investigate the association between Alpha variant and the severity of disease. The study will contribute immensely to the understanding of COVID-19 transmission dynamics by highlighting some of the risk factors for severe disease. Generally, the readership of BMJ will benefit from the study. All that is needed to implement the suggested revisions by the reviewers. Below, I indicate my proposed revisions to the manuscript. The Major revisions are number 1 through 4 while the minor revisions are split up by manuscript section.
--

Major revisions:

1. In the methods, it is best practice to identify risk factors to partake in the multivariate analysis to first screen them in a univariate analysis. Justification of this approach can be found in any statistical modelling textbook or other source. This key step is conspicuously missing in the methods section and is only mentioned once and in passing in the abstract [this one mention is the only place where “univariate” is mentioned]. This is gross error that makes me wonder whether or not, univariate was indeed performed. I need to see evidence that this analysis was actually done and its details should be a key component in the described methods and the key findings of this analysis be reported. Note that the multivariate analysis only follows from the results of the univariate analysis and thus has to be redone upon completion of the univariate analysis. Consequently, the methods can then be improved upon to capture this analysis step.
2. There may also be need to demonstrate absence of or to correct for confounding factors in the multivariate analysis as this will improve reliability of the findings.
3. The methods are generally too shallow to warrant reproduction of the statistical results. For example, it is not explicitly indicated which variables were analysed using which methods. That has rendered interpretation of the results in Table 1 difficult. Improve on the clarity and transparency of the methods.
4. The contrast/key discrepancy between the key findings of the current study and the (only) previously published study (reference #10) and perhaps the other mentioned nonpublished (local) studies warrants a more detailed discussion with reasons or at worst hypotheses on why the findings differ. It could be study assumption differences, methods-related differences and/or data span and quality differences. These need to be delved into to benefit the reader. These findings are key in the global fight against the pandemic and hence need thorough scrutiny.

Minor revisions:

Due to variations in comma usage and punctuation in general, I just mention punctuation corrections as suggestions for authors to think about and for the editorial team to deal with.

Title

Line 4 page 2. Delete the full stop at the end of the title

Abstract

Much as word count could be limiting the depth of your abstract, I feel like the issues right below are worth considering.

Line 2. Start with a generic sentence on all currently circulating variants at least in UK? Moreover, you mention the existence of B.1.351 beta variant in line 27 page 13.

Line 16. Change “was” to “were” given that data is plural.

Line 27. The second sentence should be rewritten to avoid starting it with Numerics “2341” and rather start it with words e.g. by rearranging it. You also refer to human cases as “which”, I suggest using “who” as you do it later in line 43 on page 8.

Lines 18 and 34. Ensure consistency in data writing format.

Line 37. Following my major comment above, describe here briefly how the listed factors were arrived at to partake in the multi-variate analysis. It should follow from a univariate analysis but worth explicit mention here. E.g., start with “Following their significance in the univariate analysis, obesity, age, etc were found to ... in the multivariate analysis.

Lines 46 to 48. You write “Our analysis is the first in hospitalised cohorts to show increased severity of disease associated with the Alpha variant”. From this sentence, it is not explicitly clear whether there are other analyses that found different results or that simply this analysis is the first of its kind (particularly) on UK data. Rephrase the sentence accordingly e.g., starting with “Contrary to findings from other studies, our analysis...”. Note that the existence of other studies is automatically implied in your sentence in lines 2 to 4 on page 4 as well as on page 7 in lines 33 to 36 for reference #10 study.

Strengths and limitations of this study

Line 9. Add a full stop at the end.

Ethics

Line 7. Is it “patient’s” or “patients’ ”?

Patient and public involvement

Line 14. Add a comma between need and patient?

Background

Line 8. Add date when the reported statistics were attained as the numbers are changing daily for now.

Lines 12 and 14. For clarity, add “year” because the pandemic has now spanned multiple years 2019, 2020 and 2021 so far.

Line 24. Move the full stop to after reference [4].

Methods

	Setting Line 11. Write ECMO in full. Definitions and participants Line 23. Add a comma after comparison? Determination of SARS-CoV-2 lineage Line 48. Add a comma after wave? Data sources, extraction and integration Lines 1 and 9 on page 10. Use “data were” not was? Statistical analysis and outcome measures Line 50 page 10. I suggest you use “descriptive statistics” not “general statistics” as the former is the technical term. Line 57. Add wave to read “wave one versus wave two variables” Results General epidemiology and results of viral sequencing Line 18. Rearrange to start sentence with a non-numeric. E.g. start with “Ninety one percent (1391/1528)” Same comment on line 27. Line 20. What is the rationale of using “unique”? It could be carrying some meaning that needs to be explicitly defined. Line 30. Add a comma after waves? Line 43. Replace the comma after Figure 2 with a full stop. Comparison of characteristics of admitted cases between wave one and two Line 41. Use “waves one and two” not wave in title and perhaps elsewhere in the entire manuscript that you write “wave one and two” Line 43 page 13. Use “Descriptive” instead of “general” also in the Table 1 and Table 2 headings and elsewhere in the manuscript. Line 46. You use “only a small difference”. This is not appropriate statistically speaking especially when you add in the phrase “only a small” that a statistically oriented reader would be disturbed about since the difference is statistically significant at $P=0.019$. I suggest you say “there was a statistically significant difference of 2 years between ... and ...”. That way your personal opinion is not reflected in the results.
--	--

Line 50. Delete a second full stop after).

Line 53. Add "that" after showed?

Line 57. Add a full stop after)

Table 1

Line 2 page 14. The 3rd and 4th column headers as currently indicated are not easy to follow. n is on its stand-alone row while others have percentages and IQR and medians while others have totally different quantities. Improve the presentation. Also do the same for Table 2 columns 3 and 4.

Line 35 page 15. That "Note" can better be introduced in the main Table column header with a symbol and then define the symbol at the bottom of the table. Do the same for Table 2.

Lines 4 to 9 Page 16. You write "There were small differences in other physiological parameters on admission, some of which reached statistical significance but differences were not clinically relevant." Perhaps explain more about this phrase to benefit the statistians and the clinicians at the same time. As currently written, a statistical oriented reader will be left wondering about this kind of conclusion.

Comparison of characteristics of admitted cases infected with Alpha and non-Alpha variants

Lines 48 to 53. You write "... we compared demographic, physiological and laboratory parameters between admitted cases with infection caused by Alpha variant (n=400) compared with non-Alpha (n=910) variants (Table 2)". Compared is used twice so rephrase sentence.

Line 57. Typo. Definitely not November 2021 as written.

Lines 11 to 12 page 19. You write "Cases infected with the Alpha variant were less likely to be frail (14.5% vs 22.4% p=0.001)". This is not true and neither is it what Table 2 shows.

Line 12 page 18. You use "co-morbidities" while in line 16 page 19 you use "comorbidities". Be consistent here and elsewhere in the manuscript.

Line 23 page 19. Add a comma after admission?

Comparison of non-sequenced and sequenced cases in wave two

Line 11 page 21. Use "no significant difference". Here and elsewhere, the key word is "significant". The numbers themselves could be different in magnitude (here 52.2 vs 53.8) but that difference is not statistically significant. Therefore, it is important always specify that for statistical clarity. Change that here and elsewhere applicable.

	Table 4. Replace the comma in Table 4 heading with a period Discussion Line 50 page 23. Delete [] from reference [[7]]. I miss the in-depth discussion of you results in line with those of reference #10 and other related studies. End Page 1 of 5
--	---

REVIEWER	Jassat, Waasila National Institute for Communicable Diseases, Division of Public Health Surveillance and Response
REVIEW RETURNED	03-Aug-2021

GENERAL COMMENTS	Summary: This is an important study that seeks to contribute to our understanding of the severity of new variants. Introduction: The description of emergence of new variants frames the research well. The hypothesis of the paper is sound and the objective of the study is clear. Methods: The study uses routine electronic health record data for one year at a multi-site London health care institution, comparing severity at admission among patients with alpha and non-alpha lineages. The data sources are appropriate, as well as the selection of exposure variables. The analytical methods were appropriate. The selection of patients included for analysis and the wave period could be better described. Multivariable logistic regression could be explored for comparing characteristics of patients by wave period. Discussion: The Discussion demonstrates rigor in terms of interpretation of findings and validity of conclusions drawn. It discussed prior and related work with citations. The tone and content of the conclusion and recommendations were appropriate. It would be important to explain the validity of using hypoxia as outcome. A discussion of study limitations particularly the effect of bias was missing. Major comments  - Why was hypoxia on admission the only outcome analysed? The authors should discuss the choice of this outcome and why other outcomes (ICU, ventilation, death) were not analysed. The authors should also present literature validating the use of hypoxia on admission as a marker for severity. - The wave periods were determined arbitrarily. The authors could explore a more considered choice of wave period using the national case incidence or the date when alpha was first identified. - There is some potential bias in the number of patients who were successfully sequenced. Sequenced patients in wave 2 were older, had more comorbidities including hypertension, chronic cardiac and renal diseases, which are known to be associated with more severe disease. This could bias the sequenced sample towards severity.
--

	 - Another potential bias is the possible existence of other variants which were not accounted for. - The bivariate analysis comparing the characteristics of patients in wave 1 and 2 could be strengthened using multivariate regression models. That is not the analysis of factors associated with outcome (hypoxia) but a multivariable model using variant (alpha and non-alpha) as the binary outcome variable. - The methods section should be expanded to detail patient selection more clearly. For example, the authors include in the Results "We considered all cases in wave one to be non-Alpha variants, as wave one took place prior to emergence of the Alpha variant and before Alpha variant was first identified in our population in November 2021." I would suggest a better description of these assumptions and approaches in the Methods section. - I would suggest adding short concluding remarks on the implications of the findings. Minor comments Background line 10: "estimated incidence in the first wave peaked around March 23rd at 2.2%". Clarify that this refers to incidence of new SARS-CoV-2 cases as the preceding sentence talks about deaths. Results line 46: the descriptor for the category with n=2341 is missing: "were categorised as follows, (n=2341), healthcare workers (n=1549)" Tables: should have footnotes to explain abbreviations used
--	---

REVIEWER	Atkin, Catherine University of Birmingham, Birmingham Acute Care Research, Institute of Inflammation and Ageing
REVIEW RETURNED	10-Aug-2021

GENERAL COMMENTS	This paper describes a comparison of characteristics of patients testing positive for SARS-CoV-2 at a London hospital trust. The comparison evaluates differences in demographics between two waves of disease, including age, ethnicity and comorbidity, before assessing the association between disease severity and these factors, as well as the Alpha variant as identified on sequencing. Adjusting for other factors, the Alpha variant is associated with severe disease, compared to non-variant disease. A few minor points could be addressed to improve the manuscript:  - Page 19, the formatting of the list of category definitions could be improved to aid readability - Page 19: please add a reference for the WHO ordinal scale used to define severity - Page 12, line 25: the sentence 'peaking on 28th December 2020 139 cases were diagnosed' is unclear - it would be helpful to clarify whether this was the number of cases per day? - Page 12: the sentence starting '3446/4282' appear to be an incomplete sentence, please check whether there are words missing. - Figure 1: the colours on the figure itself don't match the colour in the legend for the green/nosocomial group - Page 12, line 43: 'The' is capitalised in the middle of a sentence and should be corrected
--

	 - Page 12: (n=2341) doesn't have any explanation, this should be added. - In the second paragraph of the results section, on page 12, there are several sets of brackets comparing wave 1 and wave 2. It would be helpful to label which is wave 1 and wave 2, as it seems that this changes between different sets of brackets. Currently it is difficult to follow whether the first number within each set of brackets refers to wave 1 or wave 2. - Table 1 outlines the number of missing data for each row. It would be useful to break this down into wave 1 and wave 2, to allow the reader to assess whether missing data is balanced between the two time periods, and how this might be affecting the results. - In Table 1, NEWS2 scores contain NEWS 2 or 2+. Is the NEWS2 2+ a NEWS score of 3 or more, or does this include NEWS scores of 2. The way it is currently written is unclear. - Table 1 should include lab parameter units, if only in the legend. Some of these are included in text later, but not all. Some of the variables are for test that have more than one possible unit that could be used. - Table 2 seems to have the NEWS2 missing values in the row, which isn't consistent with the rest of the table or Table 1. - Page 24, the sentence 'the two cohorts from these hospital cohorts' isn't clear. Is this discussing two groups within two studies, or the overall cohort in two studies? - Page 26, last line: hypoxic should be hypoxia.
--	---

VERSION 1 – AUTHOR RESPONSE

Response to Reviewer 1 comments

Major revisions:

In the methods, it is best practice to identify risk factors to partake in the multivariate analysis to first screen them in a univariate analysis. Justification of this approach can be found in any statistical modelling textbook or other source. This key step is conspicuously missing in the methods section and is only mentioned once and in passing in the abstract [this one mention is the only place where “univariate” is mentioned]. This is a gross error that makes me wonder whether or not, univariate was indeed performed. I need to see evidence that this analysis was actually done and its details should be a key component in the described methods and the key findings of this analysis be reported. Note that the multivariate analysis only follows from the results of the univariate analysis and thus has to be redone upon completion of the univariate analysis. Consequently, the methods can then be improved upon to capture this analysis step.

Thank you for this important methodological point. We considered the strategy for variable selection, and referred to the TRIPOD statement (Doi: 10.7326/M14-0698). Although using univariate is quite common, that strategy (univariate analysis for selecting predictors) is “not recommended as a basis for selecting predictors, because important predictors may be rejected owing to nuances in the data set or confounding by other predictors). Thus a nonsignificant (unadjusted) statistical association with the outcome does not necessarily imply that a predictor is unimportant.” We therefore included risk factors that have been reported previously, and we included the conventional/common risk factors based on this knowledge and expert opinion. We present a literature review in the Supplementary Materials to support the variables included. Our large clinical database has many more data points than were chosen for inclusion in the model. The single mention of univariable analysis in the Abstract was an error.

2. There may also be need to demonstrate absence of or to correct for confounding factors in the multivariate analysis as this will improve reliability of the findings.

We agree that confounding factors are important for the reliability of our findings. Many of the factors included in our multivariable did not reach significance and could be considered as confounders to the outcome. There might be other confounding factors that are not considered e.g. dementia and liver disease, due to poor coding of these variables in our dataset, which is a limitation of the study. We have put this in the Discussion. Please let us know if this point requires additional clarification.

3. The methods are generally too shallow to warrant reproduction of the statistical results. For example, it is not explicitly indicated which variables were analysed using which methods. That has rendered interpretation of the results in Table 1 difficult. Improve on the clarity and transparency of the methods.

We have expanded the methods section to address this point.

4. The contrast/key discrepancy between the key findings of the current study and the (only) previously published study (reference #10) and perhaps the other mentioned nonpublished (local) studies warrants a more detailed discussion with reasons or at worst hypotheses on why the findings differ. It could be study assumption differences, methods-related differences and/or data span and quality differences. These need to be delved into to benefit the reader. These findings are key in the global fight against the pandemic and hence need thorough scrutiny.

Thank you for highlighting this area for improvement. We have updated this throughout the discussion so that it now includes the following points, especially in reference to other published studies of alpha variant severity in hospitalised patients.

- Our study outcome is prior to hospital management, and may therefore better reflect the natural history of infection prior to amelioration of severity differences by outcomes. The mortality outcome of Frampton et al [10] occurs after hospital management which may ameliorate differences in severity, as judged by mortality, of the alpha variant.
- Population level studies have often failed to control for co-morbidities (e.g. Keogh [7])
- Our data on increased females in wave two is consistent with an article in press [21] showing increased severity of the alpha variant in females

Minor revisions:

Due to variations in comma usage and punctuation in general, I just mention punctuation corrections as suggestions for authors to think about and for the editorial team to deal with.

Title Line 4 page 2. Delete the full stop at the end of the title

Actioned.

Abstract

Much as word count could be limiting the depth of your abstract, I feel like the issues right below are worth considering.

Line 2. Start with a generic sentence on all currently circulating variants at least in UK? Moreover, you mention the existence of B.1.351 beta variant in line 27 page 13.

To the Background section of the Abstract: "The Alpha variant emerged and became the dominant circulating variant in the UK in late 2020."

Line 16. Change “was” to “were” given that data is plural.

The current grammar of “The Alpha variant was first identified on 15th November 2020” appears correct to us, but we are happy to be guided by the Editors.

Line 27. The second sentence should be rewritten to avoid starting it with Numerics “2341” and rather start it with words e.g. by rearranging it.

Actioned

You also refer to human cases as “which”, I suggest using “who” as you do it later in line 43 on page 8.

When referring to cases we prefer ‘which’, and when referring to patients we opt for ‘who.’ Consistency of this choice has been checked throughout the document.

Lines 18 and 34. Ensure consistency in data writing format.

Consistency of data writing format has been checked throughout the document.

Line 37. Following my major comment above, describe here briefly how the listed factors were arrived at to partake in the multi-variate analysis. It should follow from a univariate analysis but worth explicit mention here. E.g., start with “Following their significance in the univariate analysis, obesity, age, etc were found to ... in the multivariate analysis.

Thank you, as per major comment above we have now included a literature review to justify the variables entered into the multivariable model.

Lines 46 to 48. You write “Our analysis is the first in hospitalised cohorts to show increased severity of disease associated with the Alpha variant”. From this sentence, it is not explicitly clear whether there are other analyses that found different results or that simply this analysis is the first of its kind (particularly) on UK data. Rephrase the sentence accordingly e.g., starting with “Contrary to findings from other studies, our analysis...”. Note that the existence of other studies is automatically implied in your sentence in lines 2 to 4 on page 4 as well as on page 7 in lines 33 to 36 for reference #10 study.

Thank you, this has been added.

Strengths and limitations of this study

Line 9. Add a full stop at the end.

Actioned.

Ethics

Line 7. Is it “patient’s” or “patients’ ”?

Thank you, corrected.

Patient and public involvement

Line 14. Add a comma between need and patient?

Corrected.

Background

Line 8. Add date when the reported statistics were attained as the numbers are changing daily for now.

Added, thank you.

Lines 12 and 14. For clarity, add “year” because the pandemic has now spanned multiple years 2019, 2020 and 2021 so far.

We appreciate this suggestion and have added the year where missing.

Line 24. Move the full stop to after reference [4].

Done.

Methods

Setting

Line 11. Write ECMO in full.

Done.

Definitions and participants

Line 23. Add a comma after comparison?

Done.

Determination of SARS-CoV-2 lineage

Line 48. Add a comma after wave?

Added.

Data sources, extraction and integration

Lines 1 and 9 on page 10. Use “data were” not was?

Thank you, corrected.

Statistical analysis and outcome measures

Line 50 page 10. I suggest you use “descriptive statistics” not “general statistics” as the former is the technical term.

Corrected.

Line 57. Add wave to read “wave one versus wave two variables”

Done.

Results

General epidemiology and results of viral sequencing

Line 18. Rearrange to start sentence with a non-numeric. E.g. start with “Ninety one percent (1391/1528)” Same comment on line 27.

Done.

Line 20. What is the rationale of using “unique”? It could be carrying some meaning that needs to be explicitly defined.

Unique was added to clarify that cases with more than one positive test result were only included in analysis once. This is clear in the Methods (Definitions and participants) section, so we have removed the word ‘unique’.

Line 30. Add a comma after waves?

Added.

Line 43. Replace the comma after Figure 2 with a full stop.

Corrected.

Comparison of characteristics of admitted cases between wave one and two

Line 41. Use “waves one and two” not wave in title and perhaps elsewhere in the entire manuscript that you write “wave one and two”

Corrected.

Line 43 page 13. Use “Descriptive” instead of “general” also in the Table 1 and Table 2 headings and elsewhere in the manuscript.

Corrected.

Line 46. You use “only a small difference”. This is not appropriate statistically speaking especially when you add in the phrase “only a small” that a statistically oriented reader would be disturbed about since the difference is statistically significant at $P=0.019$. I suggest you say “there was a statistically significant difference of 2 years between ... and ...”. That way your personal opinion is not reflected in the results.

Thank you for this insight, correction made.

Line 50. Delete a second full stop after).

Corrected.

Line 53. Add “that” after showed?

Added.

Line 57. Add a full stop after)

Added.

Table 1

Line 2 page 14. The 3rd and 4th column headers as currently indicated are not easy to follow. n is on its stand-alone row while others have percentages and IQR and medians while others have totally different quantities. Improve the presentation. Also do the same for Table 2 columns 3 and 4.

Thank you this has been altered, with different columns for percentages and medians.

Line 35 page 15. That "Note" can better be introduced in the main Table column header with a symbol and then define the symbol at the bottom of the table. Do the same for Table 2.

Done.

Lines 4 to 9 Page 16. You write "There were small differences in other physiological parameters on admission, some of which reached statistical significance but differences were not clinically relevant." Perhaps explain more about this phrase to benefit the statistians and the clinicians at the same time. As currently written, a statistical oriented reader will be left wondering about this kind of conclusion.

We have added two sentences to explicate our meaning: "For instance, the median heart rate was 84 (IQR 75-94) in wave one compared to 81 (IQR 72-91, $p < 0.001$), whilst significantly lower in wave two would not affect clinical interpretation of severity. Similarly a change in median mean arterial pressure of 90.7 mmHg (IQR 82.2-99.0) in wave one and 92.3 mmHg (84.7-101.3, $p < 0.001$) in wave two is similarly negligible in clinical context."

Comparison of characteristics of admitted cases infected with Alpha and non-Alpha variants

Lines 48 to 53. You write "... we compared demographic, physiological and laboratory parameters between admitted cases with infection caused by Alpha variant (n=400) compared with non-Alpha (n=910) variants (Table 2)". Compared is used twice so rephrase sentence.

Thank you, this has been amended.

Line 57. Typo. Definitely not November 2021 as written.

Thank you, corrected to November 2020.

Lines 11 to 12 page 19. You write "Cases infected with the Alpha variant were less likely to be frail (14.5% vs 22.4% $p = 0.001$).". This is not true and neither is it what Table 2 shows.

In Table 2 the proportion of individuals infected with the alpha variant with a diagnosis of frailty was 58/400 (14.5%) compared to 204/910 (22.4%, $p = 0.001$) in those not infected with the Alpha-variant. Thus we believe the above text does reflect the presented results.

Line 12 page 18. You use "co-morbidities" while in line 16 page 19 you use "comorbidities". Be consistent here and elsewhere in the manuscript.

Thank you, we have changed all instances to 'comorbidities' for consistency.

Line 23 page 19. Add a comma after admission?

Apologies, I cannot locate this missing comma.

Comparison of non-sequenced and sequenced cases in wave two x

Line 11 page 21. Use "no significant difference". Here and elsewhere, the key word is "significant". The numbers themselves could be different in magnitude (here 52.2 vs 53.8) but that difference is not statistically significant. Therefore, it is important always specify that for statistical clarity. Change that here and elsewhere applicable.

Thank you for this salient advice, which has been changed throughout the manuscript.

Table 4.

Replace the comma in Table 4 heading with a period

Thank you, punctuation has been changed.

Discussion

Line 50 page 23. Delete [] from reference [[7]].

Done

I miss the in-depth discussion of you results in line with those of reference #10 and other related studies.

Thank you, as above this has been addressed.

Response to Reviewer 2 comments

A few minor points could be addressed to improve the manuscript:

- Page 19, the formatting of the list of category definitions could be improved to aid readability

Thank you, the readability of this table has been improved by separating those categorical values represented by percentages and those continuous variables represented with median values/IQR.

- Page 19: please add a reference for the WHO ordinal scale used to define severity

Thank you, we have added a reference for the WHO ordinal scale to the Methods (page __, line __). We have also updated the Discussion to outline why this measure of severity was used (page __, line __)

- Page 12, line 25: the sentence 'peaking on 28th December 2020 139 cases were diagnosed' is unclear - it would be helpful to clarify whether this was the number of cases per day?

We have updated the text to clarify this is 139 cases 'per day'.

- Page 12: the sentence starting '3446/4282' appear to be an incomplete sentence, please check whether there are words missing.

Thank you, this has been updated. '3446/4282 (80%) of wave two cases were detected during a comparable 60 day period between 10th December 2020 and 8th February 2021.'

- Figure 1: the colours on the figure itself don't match the colour in the legend for the green/nosocomial group

Thank you, this has been corrected.

- Page 12, line 43: 'The' is capitalised in the middle of a sentence and should be corrected.

Thank you, this has been corrected.

Reviewer: 3

The selection of patients included for analysis and the wave period could be better described.

We have updated the methodology on patient inclusion (page 9, line 32):

"For the purpose of comparison only the inpatient group, admitted within 14 days following a positive test, were taken forward for onward comparison. This methodology was adopted to prevent increased testing during the pandemic affecting case ascertainment and biasing severity of cases. This is evidenced in Supplementary Figure 1, with tests increasing steadily from 100 per day to more than 1000 per day. Additionally, in wave two more interhospital transfers of severe cases requiring extracorporeal membrane oxygenation were received, mostly several days after admission. This category of patients were therefore excluded from analysis to prevent biasing towards severe disease."

"Cases with missing datapoints were dropped from multivariate analysis."

Multivariable logistic regression could be explored for comparing characteristics of patients by wave period.

Thank you for this suggestion. Comparing characteristics was done using descriptive statistics for the characteristics in table 1. We reserve multivariate logistic regression for calculating the associations between the dependent (outcome) and independent variables (variables). We would prefer not to use multivariate logistic regression for characteristics between wave one and wave as this does not contribute to our objectives/conclusions of comparing factors associated with to the outcome (hypoxia at admission as a marker of severity).

Why was hypoxia on admission the only outcome analysed? The authors should discuss the choice of this outcome and why other outcomes (ICU, ventilation, death) were not analysed. The authors should also present literature validating the use of hypoxia on admission as a marker for severity.

Thank you, we have included a justification for hypoxia as a marker of severity, and an explanation for why ICU admission and mortality was not used (page 24, line 55):

“Our finding is the first study in hospitalised cohorts to show increased severity of disease with the Alpha variant, as defined by hypoxia on admission which is equivalent to WHO ordinal scale of ≥ 4 [12] and a key marker of severe disease. Hypoxia on admission was chosen as a marker of severity to prevent confounding of results by changes in management of hospitalised patients across the pandemic. For instance treatment with steroids, which were introduced during the study period around November 2020, have been shown to reduce risk of ventilation and death [22]. Other changes in management, such as proning, anticoagulation and tocilizumab could also confound the outcomes of death and ICU admission. Hypoxia on admission is not at risk of confounding by changes in management of cases, as currently no significant management or treatment options are deployed in the community. The validity of using hypoxia as a marker of severity is shown by the clinical characteristics of SARS-COV-2, with respiratory illness causing hypoxia in a minority of cases and with a smaller proportion having respiratory failure necessitating ventilation [23].”

-The wave periods were determined arbitrarily. The authors could explore a more considered choice of wave period using the national case incidence or the date when alpha was first identified.

We chose 25th July as the separation date between waves based on local data. It was on this date our NHS Trust had the fewest cases still admitted, reflecting local incidence (page 13, line 11). Encouragingly, this similar trend is seen in national data on admissions which remained relatively static, fluctuating between around 30-80 per day between 25th July and 31st August. (<https://coronavirus.data.gov.uk/details/healthcare>). Similarly, as shown in Figure 1 our local incidence remained stably low during this time, and 25th July is relatively equidistant between the end of wave one and increase in cases seen at wave two.

There is some potential bias in the number of patients who were successfully sequenced. Sequenced patients in wave 2 were older, had more comorbidities including hypertension, chronic cardiac and renal diseases, which are known to be associated with more severe disease. This could bias the sequenced sample towards severity.

We have updated the discussion to address this (page 26, line 27):

‘We also included an assessment of bias by comparing characteristics of non-sequenced cases with those successfully sequenced. Whilst sequenced patients were older and more co-morbid there was no difference between the proportion with the outcome measure of hypoxia on admission between our sequenced and non-sequenced cases. This suggests no significant bias towards severity in the sequenced group, which was predominantly made up of cases of the Alpha variant.’

Another potential bias is the possible existence of other variants which were not accounted for.

There is little evidence for other circulating variants of concern. Despite sequencing a large proportion (1/3) of all inpatients in wave one and two only two cases of beta variant were found. This is similar to national data (<https://beta.microreact.org/project/kCk7d12Qwop1NnEeEmXvHg-uk-sars-cov-2-2020-02-052021-07-23>) which shows only small numbers of other variants of concern across the study period. For this reason we do not think it is likely that other variants could be influencing our findings.

The bivariate analysis comparing the characteristics of patients in wave 1 and 2 could be strengthened using multivariate regression models. That is not the analysis of factors associated with outcome (hypoxia) but a multivariable model using variant (alpha and non-alpha) as the binary outcome variable.

Thank you for this suggestion. We reserve multivariate logistic regression for calculating the associations between the dependent (outcome) and independent variables (variables). We would prefer not to use multivariate logistic regression for characteristics between alpha and non alpha variant as this does not contribute to our objectives/conclusions of comparing factors associated with to the outcome (hypoxia at admission as a marker of severity). If the reviewer feels strongly that this needs to be added please let us know. Our thought is that the acquisition of different variants is due to their locations or activities etc but not due to patient characteristics.

The methods section should be expanded to detail patient selection more clearly. For example, the authors include in the Results “We considered all cases in wave one to be non-Alpha variants, as wave one took place prior to emergence of the Alpha variant and before Alpha variant was first identified in our population in November 2021.” I would suggest a better description of these assumptions and approaches in the Methods section.

Our wave one cut off of 25th July 25th is 6 weeks prior to the earliest identified samples of the alpha variant on 20th Sep 2020 (<https://virological.org/t/preliminary-genomic-characterisation-of-an-emergent-sars-cov-2-lineage-in-the-uk-defined-by-a-novel-set-of-spike-mutations/563>) and 8 weeks prior to the first identification in our cohort. For this reason we feel it is a safe assumption that all wave one cases were caused by non-Alpha variants. We have moved this description into the Methods (page 11, line 14)

I would suggest adding short concluding remarks on the implications of the findings.

This has been added (page 28, line 7)

Minor comments

Background line 10: “estimated incidence in the first wave peaked around March 23rd at 2.2%”. Clarify that this refers to incidence of new SARS-CoV-2 cases as the preceding sentence talks about deaths.

Added.

Results line 46: the descriptor for the category with n=2341 is missing: “were categorised as follows, (n=2341), healthcare workers (n=1549)”

Thank you for noticing this error, which should read “Inpatients admitted within 14 days of a positive test (n=2341)”

Tables: should have footnotes to explain the abbreviations used.

Footnotes have been added to all Tables.

VERSION 2 – REVIEW

REVIEWER	Ssematimba, Amos Gulu University, Department of Mathematics, Faculty of Science
REVIEW RETURNED	18-Sep-2021

GENERAL COMMENTS	Dear Authors, I appreciated that you clearly addressed my comments. Amos
--

REVIEWER	Jassat, Waasila National Institute for Communicable Diseases, Division of Public Health Surveillance and Response
REVIEW RETURNED	23-Sep-2021

GENERAL COMMENTS	The authors have responded very well to the reviewers' comments. I am pleased to see the paper published.
---

REVIEWER	Atkin, Catherine University of Birmingham, Birmingham Acute Care Research, Institute of Inflammation and Ageing
REVIEW RETURNED	18-Sep-2021

GENERAL COMMENTS	Overall, I am happy that the authors have addressed most of the points raised previously. A small few changes are needed, mainly for clarity: Page 11, line 41: 'Cases with missing datapoints were dropped from.' Is an incomplete sentence. Page 11, line 43: 'Variables to be included in the multivariate analysis. were chosen by literature review' doesn't need a full stop in after analysis Page 12, line 16: 'to peak rapidly between the 1st and 8th April 2020 with 57 new cases.' Is unclear as to what the 57 new cases means – is this cases per day? Per week? Cases in hospital? Page 13, line 50: you've said there is a difference in the mean age of patients, but the Table states this is median (rather than mean) Page 13, line 52: 'however admitted cases were more likely to be female' – consider whether this is appropriate phrasing using however, as the phrasing suggests that the change in sex is contradictory to the change in age? Table 1: you have NEWS broken down into 0/1/2/2+ - adding the numbers, those with a NEWS of 2 are not included in the 2+ group, and therefore this category is not 2+, it is either 3+ or it is >2. 2+ implies that it includes those with a NEWS of 2 or higher, when this row appears to include only those with a NEWS over 2. Table 1: units for blood test results could be included to make this clearer – as it stands, it requires the main text to understand what units are used. Table 2: for the NEWS scores, what does the row marked 'nan' mean? Is this missing data? It is not consistent between the two tables All comments for Table 1 are also applicable to Table 2. Page 26: "Our study is limited by studying a population comes from one city" needs rephrasing to be grammatically correct
--

VERSION 2 – AUTHOR RESPONSE

Response to Reviewer 1 comments

Major revisions:

In the methods, it is best practice to identify risk factors to partake in the multivariate analysis to first screen them in a univariate analysis. Justification of this approach can be found in any statistical modelling textbook or other source. This key step is conspicuously missing in the methods section and is only mentioned once and in passing in the abstract [this one mention is the only place where “univariate” is mentioned]. This is a gross error that makes me wonder whether or not, univariate was indeed performed. I need to see evidence that this analysis was actually done and its details should be a key component in the described methods and the key findings of this analysis be reported. Note that the multivariate analysis only follows from the results of the univariate analysis and thus has to be redone upon completion of the univariate analysis. Consequently, the methods can then be improved upon to capture this analysis step.

Thank you for this important methodological point. We considered the strategy for variable selection, and referred to the TRIPOD statement (Doi: 10.7326/M14-0698). Although using univariate is quite common, that strategy (univariate analysis for selecting predictors) is “not recommended as a basis for selecting predictors, because important predictors may be rejected owing to nuances in the data set or confounding by other predictors). Thus a nonsignificant (unadjusted) statistical association with the outcome does not necessarily imply that a predictor is unimportant.” We therefore included risk factors that have been reported previously, and we included the conventional/common risk factors based on this knowledge and expert opinion. We present a literature review in the Supplementary Materials to support the variables included. Our large clinical database has many more data points than were chosen for inclusion in the model. The single mention of univariable analysis in the Abstract was an error.

2. There may also be need to demonstrate absence of or to correct for confounding factors in the multivariate analysis as this will improve reliability of the findings.

We agree that confounding factors are important for the reliability of our findings. Many of the factors included in our multivariable did not reach significance and could be considered as confounders to the outcome. There might be other confounding factors that are not considered e.g. dementia and liver disease, due to poor coding of these variables in our dataset, which is a limitation of the study. We have put this in the Discussion. Please let us know if this point requires additional clarification.

3. The methods are generally too shallow to warrant reproduction of the statistical results. For example, it is not explicitly indicated which variables were analysed using which methods. That has rendered interpretation of the results in Table 1 difficult. Improve on the clarity and transparency of the methods.

We have expanded the methods section to address this point.

4. The contrast/key discrepancy between the key findings of the current study and the (only) previously published study (reference #10) and perhaps the other mentioned nonpublished (local) studies warrants a more detailed discussion with reasons or at worst hypotheses on why the findings differ. It could be study assumption differences, methods-related differences and/or data span and quality differences. These need to be delved into to benefit the reader. These findings are key in the global fight against the pandemic and hence need thorough scrutiny.

Thank you for highlighting this area for improvement. We have updated this throughout the discussion so that it now includes the following points, especially in reference to other published studies of alpha variant severity in hospitalised patients.

- Our study outcome is prior to hospital management, and may therefore better reflect the natural history of infection prior to amelioration of severity differences by outcomes. The mortality outcome of Frampton *et al* [10] occurs after hospital management which may ameliorate differences in severity, as judged by mortality, of the alpha variant.
- Population level studies have often failed to control for co-morbidities (e.g. Keogh [7])
- Our data on increased females in wave two is consistent with an article in press [21] showing increased severity of the alpha variant in females

Minor revisions:

Due to variations in comma usage and punctuation in general, I just mention punctuation corrections as suggestions for authors to think about and for the editorial team to deal with.

Title Line 4 page 2. Delete the full stop at the end of the title

Actioned.

Abstract

Much as word count could be limiting the depth of your abstract, I feel like the issues right below are worth considering.

Line 2. Start with a generic sentence on all currently circulating variants at least in UK? Moreover, you mention the existence of B.1.351 beta variant in line 27 page 13.

To the Background section of the Abstract: "The Alpha variant emerged and became the dominant circulating variant in the UK in late 2020."

Line 16. Change "was" to "were" given that data is plural.

The current grammar of "The Alpha variant was first identified on 15th November 2020" appears correct to us, but we are happy to be guided by the Editors.

Line 27. The second sentence should be rewritten to avoid starting it with Numerics "2341" and rather start it with words e.g. by rearranging it.

Actioned

You also refer to human cases as "which", I suggest using "who" as you do it later in line 43 on page 8.

When referring to cases we prefer 'which', and when referring to patients we opt for 'who.' Consistency of this choice has been checked throughout the document.

Lines 18 and 34. Ensure consistency in data writing format.

Consistency of data writing format has been checked throughout the document.

Line 37. Following my major comment above, describe here briefly how the listed factors were arrived at to partake in the multi-variate analysis. It should follow from a univariate analysis but worth explicit mention here. E.g., start with "Following their significance in the univariate analysis, obesity, age, etc were found to ... in the multivariate analysis.

Thank you, as per major comment above we have now included a literature review to justify the variables entered into the multivariable model.

Lines 46 to 48. You write “Our analysis is the first in hospitalised cohorts to show increased severity of disease associated with the Alpha variant”. From this sentence, it is not explicitly clear whether there are other analyses that found different results or that simply this analysis is the first of its kind (particularly) on UK data. Rephrase the sentence accordingly e.g., starting with “Contrary to findings from other studies, our analysis...”. Note that the existence of other studies is automatically implied in your sentence in lines 2 to 4 on page 4 as well as on page 7 in lines 33 to 36 for reference #10 study.

Thank you, this has been added.

Strengths and limitations of this study

Line 9. Add a full stop at the end.

Actioned.

Ethics

Line 7. Is it “patient’s” or “patients’ ”?

Thank you, corrected.

Patient and public involvement

Line 14. Add a comma between need and patient?

Corrected.

Background

Line 8. Add date when the reported statistics were attained as the numbers are changing daily for now.

Added, thank you.

Lines 12 and 14. For clarity, add “year” because the pandemic has now spanned multiple years 2019, 2020 and 2021 so far.

We appreciate this suggestion and have added the year where missing.

Line 24. Move the full stop to after reference [4].

Done.

Methods

Setting

Line 11. Write ECMO in full.

Done.

Definitions and participants

Line 23. Add a comma after comparison?

Done.

Determination of SARS-CoV-2 lineage

Line 48. Add a comma after wave?

Added.

Data sources, extraction and integration

Lines 1 and 9 on page 10. Use “data were” not was?

Thank you, corrected.

Statistical analysis and outcome measures

Line 50 page 10. I suggest you use “descriptive statistics” not “general statistics” as the former is the technical term.

Corrected.

Line 57. Add wave to read “wave one versus wave two variables”

Done.

Results

General epidemiology and results of viral sequencing

Line 18. Rearrange to start sentence with a non-numeric. E.g. start with “Ninety one percent (1391/1528)” Same comment on line 27.

Done.

Line 20. What is the rationale of using “unique”? It could be carrying some meaning that needs to be explicitly defined.

Unique was added to clarify that cases with more than one positive test result were only included in analysis once. This is clear in the Methods (Definitions and participants) section, so we have removed the word ‘unique’.

Line 30. Add a comma after waves?

Added.

Line 43. Replace the comma after Figure 2 with a full stop.

Corrected.

Comparison of characteristics of admitted cases between wave one and two

Line 41. Use “waves one and two” not wave in title and perhaps elsewhere in the entire manuscript that you write “wave one and two”

Corrected.

Line 43 page 13. Use “Descriptive” instead of “general” also in the Table 1 and Table 2 headings and elsewhere in the manuscript.

Corrected.

Line 46. You use “only a small difference”. This is not appropriate statistically speaking especially when you add in the phrase “only a small” that a statistically oriented reader would be disturbed about since the difference is statistically significant at $P=0.019$. I suggest you say “there was a statistically significant difference of 2 years between ... and ...”. That way your personal opinion is not reflected in the results.

Thank you for this insight, correction made.

Line 50. Delete a second full stop after).

Corrected.

Line 53. Add “that” after showed?

Added.

Line 57. Add a full stop after)

Added.

Table 1

Line 2 page 14. The 3rd and 4th column headers as currently indicated are not easy to follow. n is on its stand-alone row while others have percentages and IQR and medians while others have totally different quantities. Improve the presentation. Also do the same for Table 2 columns 3 and 4.

Thank you this has been altered, with different columns for percentages and medians.

Line 35 page 15. That “Note” can better be introduced in the main Table column header with a symbol and then define the symbol at the bottom of the table. Do the same for Table 2.

Done.

Lines 4 to 9 Page 16. You write “There were small differences in other physiological parameters on admission, some of which reached statistical significance but differences were not clinically relevant.” Perhaps explain more about this phrase to benefit the statisticians and the clinicians at the same time. As currently written, a statistical oriented reader will be left wondering about this kind of conclusion.

We have added two sentences to explicate our meaning: “For instance, the median heart rate was 84 (IQR 75-94) in wave one compared to 81 (IQR 72-91, $p < 0.001$), whilst significantly lower in wave two would not affect clinical interpretation of severity. Similarly a change in median mean arterial pressure of 90.7 mmHg (IQR 82.2-99.0) in wave one and 92.3 mmHg (84.7-101.3, $p < 0.001$) in wave two is similarly negligible in clinical context.”

Comparison of characteristics of admitted cases infected with Alpha and non-Alpha variants

Lines 48 to 53. You write “... we compared demographic, physiological and laboratory parameters between admitted cases with infection caused by Alpha variant (n=400) compared with non-Alpha (n=910) variants (Table 2)”. Compared is used twice so rephrase sentence.

Thank you, this has been amended.

Line 57. Typo. Definitely not November 2021 as written.

Thank you, corrected to November 2020.

Lines 11 to 12 page 19. You write “Cases infected with the Alpha variant were less likely to be frail (14.5% vs 22.4% $p = 0.001$)”. This is not true and neither is it what Table 2 shows.

In Table 2 the proportion of individuals infected with the alpha variant with a diagnosis of frailty was 58/400 (14.5%) compared to 204/910 (22.4%, $p = 0.001$) in those not infected with the Alpha-variant. Thus we believe the above text does reflect the presented results.

Line 12 page 18. You use “co-morbidities” while in line 16 page 19 you use “comorbidities”. Be consistent here and elsewhere in the manuscript.

Thank you, we have changed all instances to ‘comorbidities’ for consistency.

Line 23 page 19. Add a comma after admission?

Apologies, I cannot locate this missing comma.

Comparison of non-sequenced and sequenced cases in wave two x

Line 11 page 21. Use “no significant difference”. Here and elsewhere, the key word is “significant”. The numbers themselves could be different in magnitude (here 52.2 vs 53.8) but that difference is not statistically significant. Therefore, it is important always specify that for statistical clarity. Change that here and elsewhere applicable.

Thank you for this salient advice, which has been changed throughout the manuscript.

Table 4.

Replace the comma in Table 4 heading with a period

Thank you, punctuation has been changed.

Discussion

Line 50 page 23. Delete [] from reference [[7]].

Done

I miss the in-depth discussion of you results in line with those of reference #10 and other related studies.

Thank you, as above this has been addressed.

Response to Reviewer 2 comments

A few minor points could be addressed to improve the manuscript:

- Page 19, the formatting of the list of category definitions could be improved to aid readability

Thank you, the readability of this table has been improved by separating those categorical values represented by percentages and those continuous variables represented with median values/IQR.

- Page 19: please add a reference for the WHO ordinal scale used to define severity

Thank you, we have added a reference for the WHO ordinal scale to the Methods (page __, line __). We have also updated the Discussion to outline why this measure of severity was used (page __, line __)

- Page 12, line 25: the sentence 'peaking on 28th December 2020 139 cases were diagnosed' is unclear - it would be helpful to clarify whether this was the number of cases per day?

We have updated the text to clarify this is 139 cases 'per day'.

- Page 12: the sentence starting '3446/4282' appear to be an incomplete sentence, please check whether there are words missing.

Thank you, this has been updated. '3446/4282 (80%) of wave two cases were detected during a comparable 60 day period between 10th December 2020 and 8th February 2021.'

- Figure 1: the colours on the figure itself don't match the colour in the legend for the green/nosocomial group

Thank you, this has been corrected.

- Page 12, line 43: 'The' is capitalised in the middle of a sentence and should be corrected.

Thank you, this has been corrected.

Reviewer: 3

The selection of patients included for analysis and the wave period could be better described.

We have updated the methodology on patient inclusion (page 9, line 32):

“For the purpose of comparison only the inpatient group, admitted within 14 days following a positive test, were taken forward for onward comparison. This methodology was adopted to prevent increased testing during the pandemic affecting case ascertainment and biasing severity of cases. This is evidenced in Supplementary Figure 1, with tests increasing steadily from 100 per day to more than 1000 per day. Additionally, in wave two more interhospital transfers of severe cases requiring extracorporeal membrane oxygenation were received, mostly several days after admission. This category of patients were therefore excluded from analysis to prevent biasing towards severe disease.”

“Cases with missing datapoints were dropped from multivariate analysis.”

Multivariable logistic regression could be explored for comparing characteristics of patients by wave period.

Thank you for this suggestion. Comparing characteristics was done using descriptive statistics for the characteristics in table 1. We reserve multivariate logistic regression for calculating the associations between the dependent (outcome) and independent variables (variables). We would prefer not to use multivariate logistic regression for characteristics between wave one and wave as this does not contribute to our objectives/conclusions of comparing factors associated with to the outcome (hypoxia at admission as a marker of severity).

Why was hypoxia on admission the only outcome analysed? The authors should discuss the choice of this outcome and why other outcomes (ICU, ventilation, death) were not analysed. The authors should also present literature validating the use of hypoxia on admission as a marker for severity.

Thank you, we have included a justification for hypoxia as a marker of severity, and an explanation for why ICU admission and mortality was not used (page 24, line 55):

“Our finding is the first study in hospitalised cohorts to show increased severity of disease with the Alpha variant, as defined by hypoxia on admission which is equivalent to WHO ordinal scale of ≥ 4 [12] and a key marker of severe disease. Hypoxia on admission was chosen as a marker of severity to prevent confounding of results by changes in management of hospitalised patients across the pandemic. For instance treatment with steroids, which were introduced during the study period around November 2020, have been shown to reduce risk of ventilation and death [22]. Other changes in management, such as proning, anticoagulation and tocilizumab could also confound the outcomes of death and ICU admission. Hypoxia on admission is not at risk of confounding by changes in management of cases, as currently no significant management or treatment options are deployed in the community. The validity of using hypoxia as a marker of severity is shown by the clinical characteristics of SARS-COV-2, with respiratory illness causing hypoxia in a minority of cases and with a smaller proportion having respiratory failure necessitating ventilation [23].”

-The wave periods were determined arbitrarily. The authors could explore a more considered choice of wave period using the national case incidence or the date when alpha was first identified.

We chose 25th July as the separation date between waves based on local data. It was on this date our NHS Trust had the fewest cases still admitted, reflecting local incidence (page 13, line 11). Encouragingly, this similar trend is seen in national data on admissions which remained relatively static, fluctuating between around 30-80 per day between 25th July and 31st August. (<https://coronavirus.data.gov.uk/details/healthcare>). Similarly, as shown in Figure 1 our local incidence remained stably low during this time, and 25th July is relatively equidistant between the end of wave one and increase in cases seen at wave two.

There is some potential bias in the number of patients who were successfully sequenced. Sequenced patients in wave 2 were older, had more comorbidities including hypertension, chronic cardiac and renal diseases, which are known to be associated with more severe disease. This could bias the sequenced sample towards severity.

We have updated the discussion to address this (page 26, line 27):

'We also included an assessment of bias by comparing characteristics of non-sequenced cases with those successfully sequenced. Whilst sequenced patients were older and more co-morbid there was no difference between the proportion with the outcome measure of hypoxia on admission between our sequenced and non-sequenced cases. This suggests no significant bias towards severity in the sequenced group, which was predominantly made up of cases of the Alpha variant.'

Another potential bias is the possible existence of other variants which were not accounted for.

There is little evidence for other circulating variants of concern. Despite sequencing a large proportion (1/3) of all inpatients in wave one and two only two cases of beta variant were found. This is similar to national data (<https://beta.microreact.org/project/kCk7d12Qwop1NnEeEmXvHq-uk-sars-cov-2-2020-02-052021-07-23>) which shows only small numbers of other variants of concern across the study period. For this reason we do not think it is likely that other variants could be influencing our findings.

The bivariate analysis comparing the characteristics of patients in wave 1 and 2 could be strengthened using multivariate regression models. That is not the analysis of factors associated with outcome (hypoxia) but a multivariable model using variant (alpha and non-alpha) as the binary outcome variable.

Thank you for this suggestion. We reserve multivariate logistic regression for calculating the associations between the dependent (outcome) and independent variables (variables). We would prefer not to use multivariate logistic regression for characteristics between alpha and non alpha variant as this does not contribute to our objectives/conclusions of comparing factors associated with to the outcome (hypoxia at admission as a marker of severity). If the reviewer feels strongly that this needs to be added please let us know. Our thought is that the acquisition of different variants is due to their locations or activities etc but not due to patient characteristics.

The methods section should be expanded to detail patient selection more clearly. For example, the authors include in the Results “We considered all cases in wave one to be non-Alpha variants, as wave one took place prior to emergence of the Alpha variant and before Alpha variant was first identified in our population in November 2021.” I would suggest a better description of these assumptions and approaches in the Methods section.

Our wave one cut off of 25th July 25th is 6 weeks prior to the earliest identified samples of the alpha variant on 20th Sep 2020 (<https://virological.org/t/preliminary-genomic-characterisation-of-an-emergent-sars-cov-2-lineage-in-the-uk-defined-by-a-novel-set-of-spike-mutations/563>) and 8 weeks prior to the first identification in our cohort. For this reason we feel it is a safe assumption that all wave one cases were caused by non-Alpha variants. We have moved this description into the Methods (page 11, line 14)

I would suggest adding short concluding remarks on the implications of the findings.

This has been added (page 28, line 7)

Minor comments

Background line 10: “estimated incidence in the first wave peaked around March 23rd at 2.2%”. Clarify that this refers to incidence of new SARS-CoV-2 cases as the preceding sentence talks about deaths.

Added.

Results line 46: the descriptor for the category with n=2341 is missing: “were categorised as follows, (n=2341), healthcare workers (n=1549)”

Thank you for noticing this error, which should read “Inpatients admitted within 14 days of a positive test (n=2341)”

Tables: should have footnotes to explain the abbreviations used.

Footnotes have been added to all Tables.

VERSION 3 – REVIEW

REVIEWER	Atkin, Catherine University of Birmingham, Birmingham Acute Care Research, Institute of Inflammation and Ageing
REVIEW RETURNED	07-Oct-2021
GENERAL COMMENTS	Thank you for addressing the previous comments - I am happy that all previously raised points have been addressed.